# iNOS is necessary for GBP-mediated *T. gondii* clearance in murine macrophages via vacuole nitration and intravacuolar network collapse

Xiao-Yu Zhao [1], Samantha L. Lempke[1], Jan C. Urbán Arroyo [1], Isabel G. Brown[1], Bocheng Yin[1], Magdalena M. Magaj [2], Nadia K. Holness[1], Jamison Smiley[1], Stefanie Redemann [2] & Sarah E. Ewald [1] ✉

*Toxoplasma gondii* is an obligate intracellular parasite of rodents and humans. Interferon-inducible guanylate binding proteins (GBPs) are mediators of *T. gondii* clearance, however, this mechanism is incomplete. Here, using automated spatially targeted optical micro proteomics we demonstrate that inducible nitric oxide synthetase (iNOS) is highly enriched at GBP2$^+$ parasitophorous vacuoles (PV) in murine macrophages. iNOS expression in macrophages is necessary to limit *T. gondii* load in vivo and in vitro. Although iNOS activity is dispensable for GBP2 recruitment and PV membrane ruffling; parasites can replicate, egress and shed GBP2 when iNOS is inhibited. *T. gondii* clearance by iNOS requires nitric oxide, leading to nitration of the PV and collapse of the intravacuolar network of membranes in a chromosome 3 GBP-dependent manner. We conclude that reactive nitrogen species generated by iNOS cooperate with GBPs to target distinct structures in the PV that are necessary for optimal parasite clearance in macrophages.

*T oxoplasma gondii* is an obligate intracellular parasite with a remarkably broad intermediate host range that includes humans and mice. Although *T. gondii* infects approximately one-third of the human population, the infection is often asymptomatic[1]. Severe complications emerge in immuno-compromised individuals or during infection of the fetus, underscoring the importance of a functional immune response to control *T. gondii* infection[2].

The parasite's ability to infect and grow within most nucleated cell types in hundreds of host species is related to the formation and maintenance of the parasitophorous vacuole (PV). The PV membrane (PVM) is formed from the host plasma membrane during *T. gondii* invasion and connected to the parasite by an intra-vacuolar network (IVN) of host-derived lipid nano-tubes[3]. The parasite uses secreted effectors to maintain the PVM and the IVN membranes, recruit

nutrients to the vacuole, and evade immune recognition[4]. Formation of the PV facilitates evasion from Toll-like receptors (TLR) 7, −8, −9[5,6] and −11 (TLR11 is a pseudogene in humans), which recognize phagocytosed, damaged parasites (Fig. 1a)[7]. Disruption of the gut epithelium during oral infection leads to TLR2, −4, and −9 signaling in response to commensal microbiota[5,8,9]. TLR signaling through NF-κB produces interleukin 12 (IL-12), which is critical for IFNγ production[7,10,11]. IFNγ is necessary for both the protective CD8 T cell response and for cell-autonomous control of parasite growth in human and murine cells[12].

IFNγ receptor signaling through STAT1 transcriptionally upregulates interferon-inducible GTPases (IIGs) which survey cells for damaged or non-self-membranes and target them for removal (Fig. 1a)[13–15]. In mice, the p47 immunity-related GTPases (IRG)[16], *Irgb6*[17], *Irga6*[18] are recruited to the PVM[19,20], under the regulatory control of

¹Department of Microbiology, Immunology, and Cancer Biology at the Carter Immunology Center, University of Virginia School of Medicine, Charlottesville, VA, USA. ²Center for Membrane and Cell Physiology, Department of Molecular Physiology and Biological Physics, University of Virginia School of Medicine, Charlottesville, VA, USA. ✉e-mail: se2s@virginia.edu

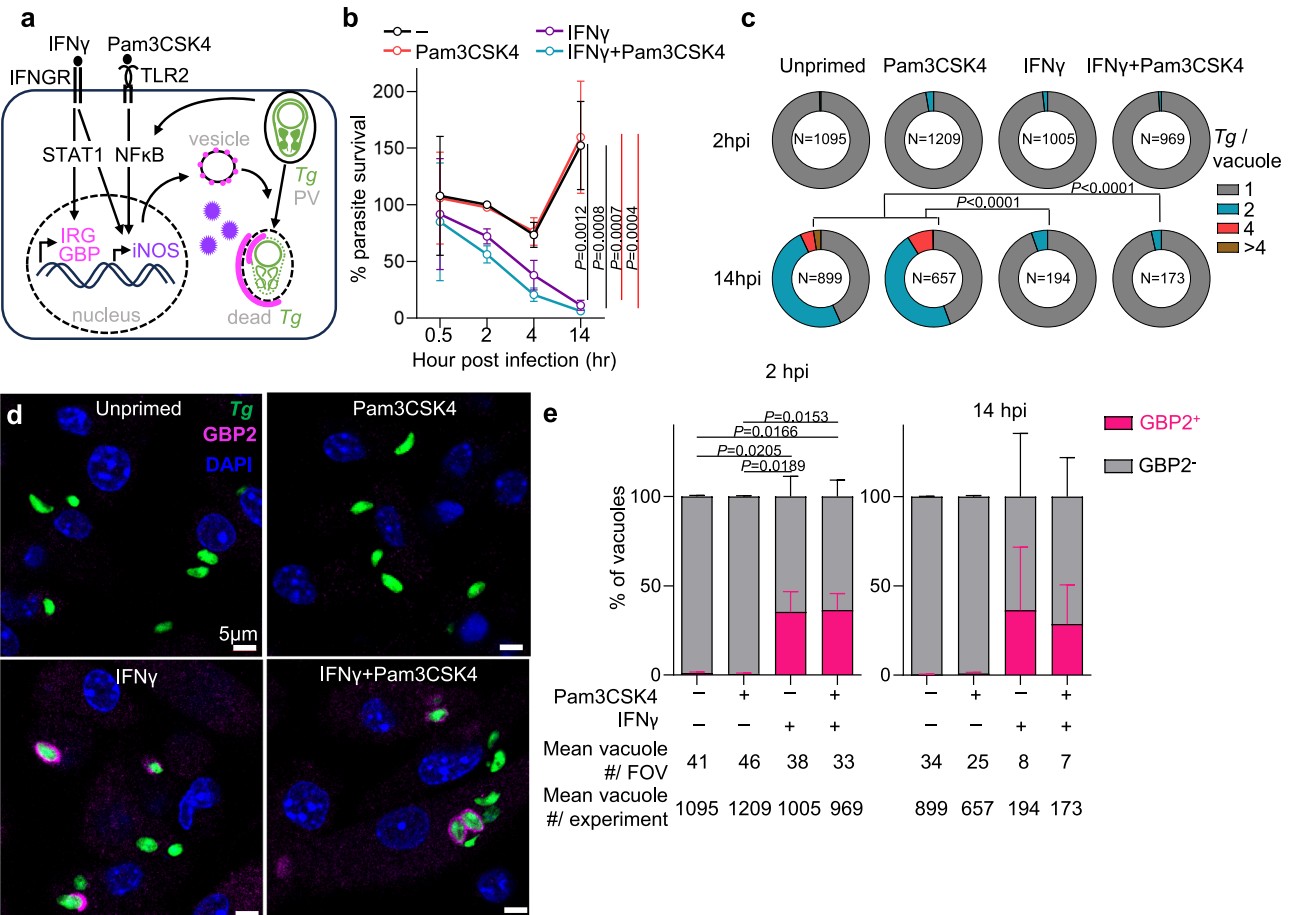

**Fig. 1 | IFNγ stimulation leads to heterogenous GBP2 recruitment to the *T. gondii* vacuole in BM-MoDCs. a** Schematic of cell-autonomous parasite clearance. IRG and GBP proteins (magenta) are induced by STAT1 downstream of IFN-γ signaling through the IFN-γ receptor (IFNGR). iNOS (purple) is transcriptionally co-regulated by STAT1 and NFκB downstream of Toll-like receptor (TLR) signaling and/or parasite effectors. IRG and GBPs are stored in vesicles and oligomerize on the parasitophorous vacuole (PV) and parasite plasma membranes contributing to *T. gondii* killing. iNOS is expressed in the cytosol and uses interaction partners to associate with membranes. **b–e** B6 mouse bone marrow monocyte/dendritic cells (BM-MoDCs) differentiated with GM-CSF were pretreated for 20 h with media, 70 ng/ml IFNγ and/or 100 ng/ml of Pam3CSK4 for the final 3 h before infection. BM-

MoDCs were infected with *T. gondii* expressing GFP and luciferase (Me49GFP-Luc *Tg*) at MOI 1.5. **b**, At 0.5, 2, 4, or 14-h post-infection (hpi) parasite load was quantified by luciferase assay and plotted as relative light units (RLU) normalized to unstimulated, infected cells at 2 hpi. $N = 3$ independent experiments. **c** The number of *Tg* per vacuole (1, 2, 4 or greater than 4 parasites) was quantified by microscopy and plotted as the proportion of all vacuoles. **d** GBP2 (magenta) localization to the *Tg* vacuole (green) was determined by immunofluorescence microscopy, counterstained with DAPI (blue). **e** The number of GBP2 positive vacuoles (magenta) was quantified and plotted as the percent of all vacuoles. $N = 3$ independent experiments. Scale bar, 5 μm. Error bars represent Mean ± SEM by one-way ANOVA (**b**) or 2-way ANOVA (**c, e**) with Tukey post hoc test.

IRGM1, −2 and −3[16,21–23]. In addition, the p65 small guanylate-binding proteins (GBPs) encoded on chromosome 3 (GBP1–3, 5, and 7)[24] and chromosome 5 (GBPs 6, 9)[25,26] are recruited to PVM and play partially redundant roles in vacuole disruption and parasite clearance in cell culture and in vivo[24,27–29]. Live imaging studies have shown that IRGb6 recruitment to the PV is followed by the re-localization of fluorescent proteins from the host cytosol to the parasite then parasite rounding (a measure of death)[19]. Mouse IRGb6[17], GBP1[27] and chromosome 3 GBPs[24] are associated with PVM 'ruffling' and discontinuity by transmission electron microscopy. Human GBP1 (the homolog of mouse GBP2) positive vacuoles are associated with permeability to cytosolic dye and leakage of parasite nucleic acids to the cytosol compared to GBP1-negative vacuoles[30]. These data indicate that IIGs are necessary for vacuole targeting and permeability; however, they have not been shown to be sufficient for *T. gondii* killing, indicating that there are gaps in the mechanistic understanding of this process.

Heterogeneous IIG targeting to the PV has been a major challenge to understanding the mechanism of cell-autonomous *T. gondii* clearance. The frequency of IIG-positive vacuoles ranges from 30–60% of all PV depending on the identity of the IIG, host and parasite cell type[24,25,27,28,30]. Tools to isolate the PV are limited. Subcellular fractionation does not preserve the intact PVM[31]. A proximity biotinylation tool to study the PV proteome has been developed, however, it cannot differentiate between IIG-targeted and IIG-negative vacuoles in the population[32]. An additional hurdle is that IRG and GBP activity is also spatially regulated. A single host cell contains both inactive pools of the GTPases localized to vesicles, Golgi, or ER and active GTPase pools targeted to pathogen-associated membranes or damaged-self membranes[14,15].

Here, we leverage the heterogeneity in IIG recruitment to the PV for comparative proteomics using automated spatially targeted optical micro-proteomics (autoSTOMP)[33–35]. This led to the discovery that iNOS expression in murine macrophages and dendritic cells is necessary for efficient parasite clearance by GBPs recruited to the PVM. iNOS activation leads to nitration of the vacuole by reactive nitrogen species, causing the collapse of the inner vacuolar network (IVN) which is necessary to evade antigen presentation and may regulate nutrient import from the host and transport of parasite effectors to the host cytosol[3,36,37]. The efficiency of IVN collapse and iNOS-mediated killing is dependent on the chromosome 3 GBPs, and parasites can evade GBP-

clearance in the absence of iNOS, indicating that these two innate immune effector pathways collaborate to mediate vacuole-autonomous parasite clearance in murine myeloid cells.

## Results

### GBP2 recruitment to the *T. gondii* PVM is heterogeneous in mouse myeloid cells

To assess the role of immunological stress on *T. gondii* clearance, bone marrow derived monocytes and dendritic cells (BM-MoDCs) were incubated with media alone, IFNγ and/or the TLR2 ligand Pam3CSK4 (Fig. 1a). BM-MoDCs were then infected with the type II *T. gondii* strain Me49 engineered to express GFP and luciferase. BM-MoDCs treated with IFNγ alone or IFNγ and Pam3CSK4 efficiently restricted *T. gondii* growth by 14 h post-infection (Fig. 1b); however, TLR2 stimulation alone was not sufficient for parasite clearance. The IFNγ-dependent clearance corresponded with decreased number of PV and parasites per vacuole by 14 h post-infection (hpi) (Fig. 1c).

To measure parasite vacuole detection by interferon-inducible GTPases, samples were stained with an antibody raised against mouse GBP2. GBP2 was selected because there is evidence that GBP targeting is downstream of p47 IRG recruitment to the vacuole[17], GBP2 is the homolog of human GBP1, and GBP2 has been shown to interact with both the parasite vacuole membrane[28] and the parasite following vacuole permeabilization[26]. GBP2 co-localized with 40% of *T. gondii* vacuoles by two hours post-infection in BM-MoDCs treated with IFNγ or IFNγ and Pam3CSK4 (Fig. 1d), but not in unstimulated or Pam3CSK4-only stimulated conditions. Cytosolic puncta of GBP2 were also observed, consistent with the presence of a non-targeted, vesicular pool of GBP2[25,26,38]. While the total number of parasites decreased over time (Fig. 1b, c) the proportion of GBP2+ and GBP2- vacuoles was similar at 2- and 14-hpi (Fig. 1e). This heterogeneity in GBP targeting is consistent with previous observations in mouse macrophages[27] and embryonic fibroblasts (MEFs)[28].

### iNOS is enriched near parasite vacuoles in IFNγ-treated BM-MoDCs by autoSTOMP

IIG biology is regulated transcriptionally and post-translationally via localization, phosphorylation and prenylation[13,14,39–42]. We expected host effectors regulating parasite clearance to be differentially enriched on PV under IFNγ-treated conditions compared to non-restrictive conditions. To identify regulators of IIG-mediated parasite clearance we used automated spatially targeted optical micro proteomics (autoSTOMP)[33–35] to image and selectively photo-biotinylate proteins in the region surrounding the *T. gondii* signal in unstimulated, Pam3CSK4, IFNγ or IFNγ and Pam3CSK4 stimulated BM-MoDCs (Fig. 2a, b, PVM). On replicate coverslips, autoSTOMP was used to selectively biotinylate regions where GBP2 colocalized with the *T. gondii* signal in the IFNγ or IFNγ and Pam3CSK4 stimulated conditions (Fig. 2a, GBP2+). Biotinylated proteins were streptavidin precipitated and identified by liquid chromatography-mass spectrometry (LC-MS) with label-free quantification.

2,070 host proteins were identified across the six PVM and GBP2+ samples (Fig. 2b, Supplementary Data 1). Of these, only 52 proteins were similarly enriched in negative controls where photo-crosslinking did not occur (Supplementary Data 1, Dark controls). Principal component analysis confirmed biological replicates were more similar to each other than differentially stimulated conditions collected and analyzed on the same days (Fig. 2c). Of note, the GBP2+ 'IFNγ' and 'IFNγ and Pam3CSK4' stimulated conditions clustered tightly (Fig. 2d pink, blue). This indicates that IIG positive vacuole region has a conserved protein signature, independent of TLR2 stimulation. Peptides that did not align to the murine protein library were searched against a *T. gondii* protein library[33], which identified 86 parasite proteins enriched across the six samples (Supplementary Fig. 1). These included rhoptry protein (ROP)7 and

ROP5, which localize to the cytoplasmic face of the vacuole after being secreted into the host cytosol[43].

As anticipated, several IIGs were highly enriched in PVM and GBP2+ samples under 'IFNγ' and 'IFNγ and Pam3CSK4' stimulation (Fig. 2d, e). These included IIGP1 (IRGa6)[20], TGTP2 (IRGb6)[17], IRGM2[21], GBP1, GBP2, GBP5, and GBP7 which have been shown to localize to the PVM[19,20,25,26]. In addition, GM4951, a predicted IFN-inducible GTPase, and GVIN1, a very large IFN-inducible GTPase, whose function in *T. gondii* infection has not been explored, were also identified in proximity to the vacuole. Few proteins were significantly differentially enriched when PVM and GBP2+ samples were compared within the 'IFNγ' or the 'IFNγ and Pam3CSK4' stimulated conditions (Fig. 2e). Since the resolution of autoSTOMP (~1μm) is wider than the parasite vacuole membrane structure, we sought to determine if sampling of near neighbor regions of the cell could be contributing to this signature by dilating the PVM Map by an additional pixel (~1μm) and identifying proteins in this region (Supplementary Data 1, Supplementary Fig. 2). Although there was a shift towards greater enrichment of GBPs and IRGs in the IFNγ-GBP2 + PVM vs. dilate sample compared to IFNγ-PVM, they were not significantly differentially enriched. Notably, both samples contained some significantly differentially enriched proteins. Thus, we interpret these data to mean that the resolution of autoSTOMP in these assays is ~2 μm and that the technique is identifying a mixture of vacuole-associated proteins and high abundance, near neighbor proteins in the infected cell.

Inducible nitric oxide synthase (iNOS, encoded by the gene *Nos2*) was among the most enriched proteins in the PVM and GBP2+ regions of IFNγ and Pam3CSK4 stimulated condition (Fig. 2d, e, purple). iNOS was also enriched in the IFNγ-stimulated GBP2+ region relative to the unprimed condition and the IFNγ PVM condition, although this was not statistically significant due to variation in the replicates, and potentially, abundant expression in the cytosol (Fig. 2d, e). Nitric oxide synthetases (endothelial eNOS, neuronal nNOS, and iNOS) produce nitric oxide (·NO) a membrane-diffusible, radical gas that regulates cell signaling[44]. iNOS is unique in its capacity for transcriptional upregulation in myeloid cells leading to millimolar concentrations of ·NO and conversion to microbicidal reactive species[45].

### iNOS in myeloid cells is necessary to survive *T. gondii* infection

An essential role for iNOS in host resistance to *T. gondii* was first determined in 1997 using whole-body knockout mice[46,47]. To determine if iNOS expression in myeloid cells was necessary to protect the host from *T. gondii* in vivo we generated myeloid specific *Nos2* knockouts (*LysM^{Cre/wt}Nos2^{fl/fl}*). Infected *LysM^{Cre}Nos2^{fl/fl}* mice met euthanasia requirements (Fig. 3a, pink), with delayed kinetics compared to whole body knockouts (Fig. 3a, violet), exhibiting more extreme weight loss during acute infection than *LysM^{wt}Nos2^{fl/fl}* controls (Fig. 3b, gray).

To evaluate iNOS-dependent changes in immune infiltrate, peritoneal exudate cells (PECs) were isolated for flow cytometry 9 days post-infection (dpi), before *Nos2^{-/-}* mice became moribund. Compared to *LysM^{wt}Nos2^{fl/fl}* mice, *Nos2^{-/-}* and *LysM^{Cre}Nos2^{fl/fl}* mice had a significant reduction in Ly6C+ macrophages as well as a compensatory increase in small, panel-negative cells, which were likely lymphocytes (Fig. 3c, Supplementary Fig. 3). Approximately 30% of CD45+ PECs were iNOS-positive in *LysM^{wt}Nos2^{fl/fl}* mice the majority of which were Ly6C+ macrophages (Fig. 3d). In *LysM^{Cre}Nos2^{fl/fl}* PECs there was a significant reduction in iNOS+Ly6C+ macrophages. Although the number of iNOS-positive cells varied from mouse to mouse, the mean fluorescence intensity (MFI) of iNOS was significantly lower in the iNOS+*LysM^{Cre}Nos2^{fl/fl}* Ly6C+macrophages, Ly6C− macrophages and dendritic cell populations than *LysM^{wt}Nos2^{fl/f}* mice (Fig. 3e), suggesting that iNOS may be upregulated in the myeloid progenitors before the LysM promoter is activated, resulting in residual levels of iNOS proteins in the differentiated cells[48]. LysM-cre has been reported to drive

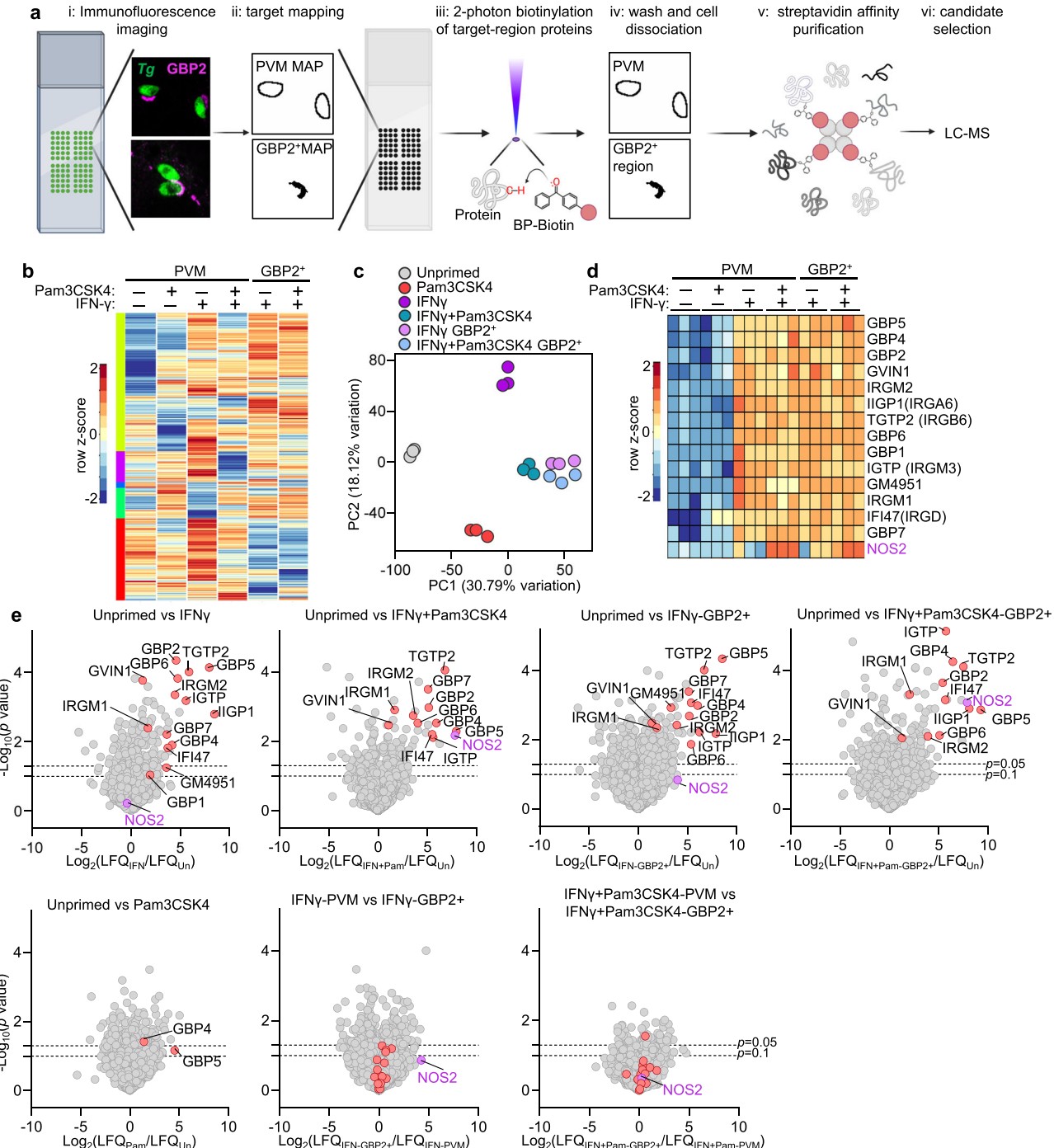

**Fig. 2 | AutoSTOMP identifies inducible nitric oxide synthase (iNOS) enriched near parasite vacuoles in IFNγ and Pam3CSK4-treated BM-MoDCs. a** BM-MoDCs were prepared for autoSTOMP as described in Fig. 1. i, At 2 hpi cells were fixed and stained for immunofluorescence imaging with *Tg* and GBP2-specific antibodies. ii, Images were used to identify the pixel coordinates of the regions containing the parasite vacuole membrane and immediate cytosol (PVM MAP) or regions where GBP2 co-localized with the vacuole (GBP2⁺ MAP). iii, MAP files were used to guide 2-photon laser excitation to photo-crosslinking BP-biotin to any protein in the target regions. (iv) Unconjugated BP-biotin was washed away and cell lysates were prepared for (v) streptavidin bead affinity purification. vi, biotinylated proteins were identified by LC-MS and label-free quantification (LFQ). **b** 2070 mouse

proteins were identified across 6 autoSTOMP conditions. Heatmap represents the row Z-score of each protein's log₂LFQ, averaged across $N = 3$ biological replicates per condition, clustered by Z-score similarity across conditions (color blocks at left). **c** PCA analysis of autoSTOMP replicates. **d** IFN-inducible GTPases (IIGs) and iNOS identified in autoSTOMP, cropped heat map from (**b**). **e** Pairwise comparison of proteins enriched in the unprimed PVM region (left, negative values) relative to Pam3CSK4, IFNγ, IFNγ+Pam3CSK4 PVM or the subset of GBP2⁺ vacuoles; IFNγ PVM vs. IFNγ GBP2⁺ vacuoles or IFNγ+Pam3CSK4 PVM vs. IFNγ+Pam3CSK4 GBP2⁺ vacuoles. IIGs, red; iNOS, violet. Plots showing −log₁₀(*p* value) from two-sided Student's *t* test and log₂LFQ differences between each comparison. Dotted lines indicate $p = 0.1$ and $p = 0.05$, respectively.

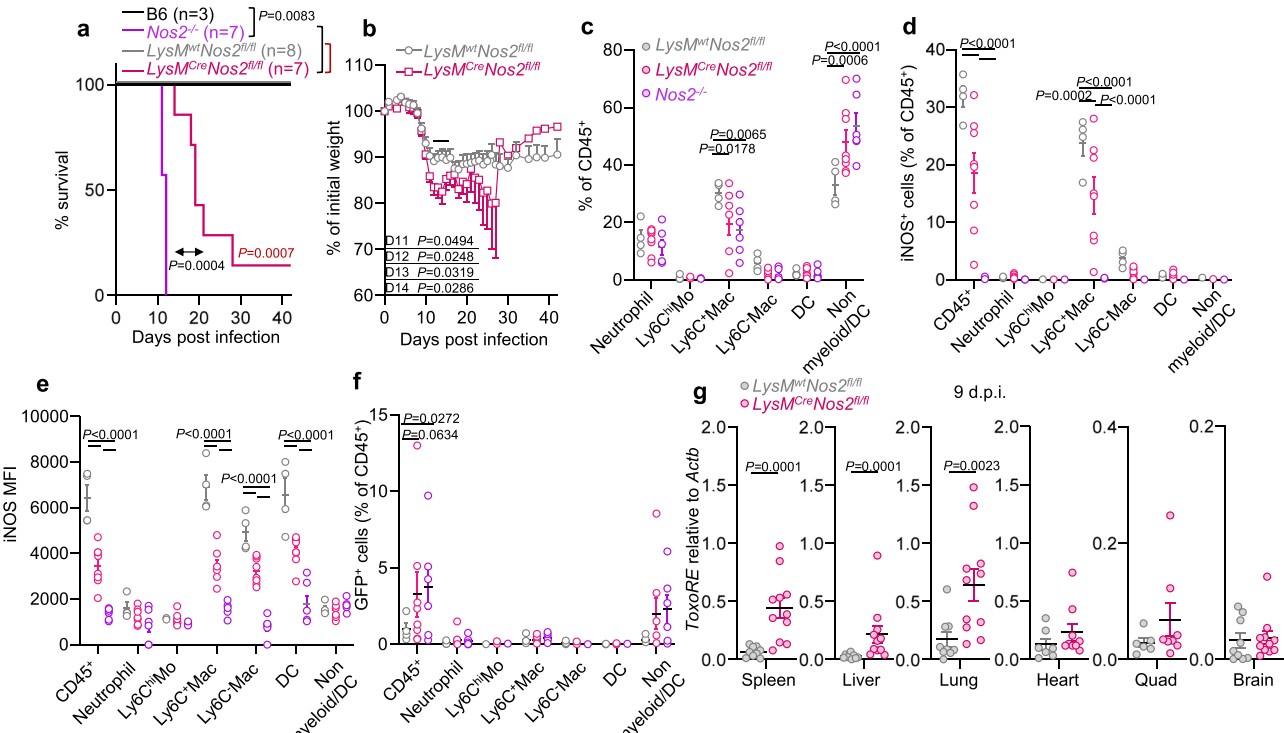

**Fig. 3 | iNOS expression on myeloid cells is necessary for *T. gondii* restriction in vivo and host survival. a–g** Age and sex-matched B6 (black) and *Nos2⁻/⁻* (violet) mice or *LysM^wt Nos2^fl/fl* (gray) and *LysM^Cre Nos2^fl/fl* (magenta) littermates were intra-peritoneally injected with 5 Me49-GFP-Luc cyst. Euthanasia criteria (**a**, Log-rank test) and weight loss (**b**, two-sided Student's *t* test) were monitored for 42 days post-infection (dpi). **c–f** 8-9 d.p.i. peritoneal exudate was collected and CD45⁺ cells were evaluated by flow cytometry. Neutrophil (CD11b⁺F4/80⁻Ly6G⁺), Ly6C^hi monocytes (Ly6C^hi Mo, CD11b⁺F4/80⁻Ly6G⁻IA/IE⁻Ly6C^hi), Ly6C⁺ macrophage (Ly6C⁺ Mac, CD11b⁺F4/80⁺Ly6G⁻IA/IE⁻Ly6C⁺), Ly6C⁻ macrophage (Ly6C⁻ Mac, CD11b⁺F4/80⁺Ly6G⁻IA/IE⁺Ly6C⁻), dendritic cells (DC, CD11c⁺IA/IE⁺), non-myeloid or DC (non-myeloid/DC, CD11b⁻CD11c⁻Ly6G⁻F4/80⁻) were analyzed from *Nos2⁻/⁻* (*N* = 6), *LysM^wt Nos2^fl/fl* (*N* = 4) and *LysM^Cre Nos2^fl/fl* (magenta *N* = 6) animals. **d–e** For each population, iNOS-positive cells were gated (**d**) and iNOS mean fluorescence intensity (MFI) was determined (**e**). **f** For each population, *Tg*-GFP positive cells were gated. **g** Parasite burden in the spleen, liver, lung, heart, skeletal muscle (Quadricep, Quad), and brain genomic DNA were analyzed by qPCR using primer probes specific to *ToxoRE* normalized to mouse *Actb*, *N* = 9 *LysM^wt Nos2^fl/fl* and *N* = 11 *LysM^Cre Nos2^fl/fl*, two-tailed Mann–Whitney test. **b–g** Error bars represent Mean ± SEM. Only half of the error bars were shown in (**b**) for clarity.

gene deletion in neutrophils, however, iNOS expression was not observed in neutrophils, Ly6C^hi inflammatory monocytes, or the panel negative cells relative to *Nos2⁻/⁻* controls (Fig. 3d, e).

Parasite levels are low in the peritoneal cavity by 9 dpi, nevertheless, there was a significant increase in GFP⁺ (*T. gondii*) CD45⁺ cells in the *Nos2⁻/⁻* compared to the *LysM^wt Nos2^fl/fl* (Fig. 3f). *T. gondii* load was significantly higher in the spleen, liver, and lung of *LysM^Cre Nos2^fl/fl* compared to *LysM^wt Nos2^fl/f* littermates (Fig. 3g), sites of dissemination from the peritoneal cavity. There was a trend toward higher parasite levels in the skeletal muscle, heart, and brain, however, this was not significant, which may reflect the early stage of parasite colonization of these tissues. Based on these data we conclude that optimal iNOS expression in the macrophage and monocyte lineages is necessary to control *T. gondii* burden and host survival following intra-peritoneal infection.

## iNOS activity is necessary for chromosome 3 GBP-mediated *T. gondii* clearance

The in vivo requirement for iNOS in the myeloid cells and the proximity of iNOS to GBP2⁺ PVs led us to hypothesize that iNOS is necessary for cell-autonomous clearance of *T. gondii* in GBP-targeted vacuoles. Consistent with the literature, IFNγ stimulation upregulated *Nos2* transcript (Fig. 4a) and protein levels (Fig. 4b) in a way that was significantly enhanced by TLR stimulation[49]. Notably, *T. gondii* infection was sufficient to synergize with IFNγ to upregulate iNOS, which is likely due to NF-κB phosphorylation by the parasite effectors GRA15[50], GRA24[51,52] and/or parasite ligation of TLRs[7]. By comparison, *Nos1* and

*Nos3* transcripts were not detected in BM-MoDCs (Supplementary Fig. 4a).

To determine if iNOS was necessary for IFNγ-mediated *T. gondii* clearance, we infected BM-MoDCs from B6 or *Nos2⁻/⁻* animals. INOS deficiency rescued *T. gondii* growth in BM-MoDCs treated with IFNγ or IFNγ and Pam3CSK4 to levels that were statistically similar to unstimulated BM-MoDCs (Fig. 4c). Blocking iNOS activity with the selective iNOS inhibitor 1400W (Fig. 4d) rescued parasite growth as efficiently as deleting *Nos2* (Fig. 4c). In addition, we confirmed that *Nos2*-deletion did not rescue parasite growth by negatively impacting the transcript expression of chromosome 3 GBPs, effector IRGb6 and IRGb10, or GBP2 protein levels in response to IFNγ (Supplementary Fig. 4b, c).

To determine if IIGs were necessary for iNOS-mediated parasite clearance we interrupted *Nos2* (RAW^ΔNos2) or deleted the chromosome 3 GBPs (RAW^ΔGbp-chr3) in RAW 264.7 macrophages using CRISPR/Cas9 (Supplementary Fig. 5a–c). iNOS was necessary for IFNγ-mediated parasite clearance in RAW 264.7 macrophages (Fig. 4e, violet). Deleting chromosome 3 GBPs rescued parasite growth in IFNγ-treated RAWs to ~30% of unstimulated levels (Fig. 4e, red vs. black). This degree of rescue was consistent with previous studies in macrophages, but less efficient than iNOS deletion, most likely due to the functional redundancy of chromosome 5 GBPs and/or IRGs[24]. Importantly, treating IFNγ-stimulated RAW^ΔGbp-chr3 cells with 1400W did not enhance parasite growth beyond 1400W treatment alone, consistent with the interpretation that iNOS and the chromosome 3 GBPs function in the same pathway rather than additively to promote parasite clearance. IFNγ and TLR stimulation induces endosome maturation which could

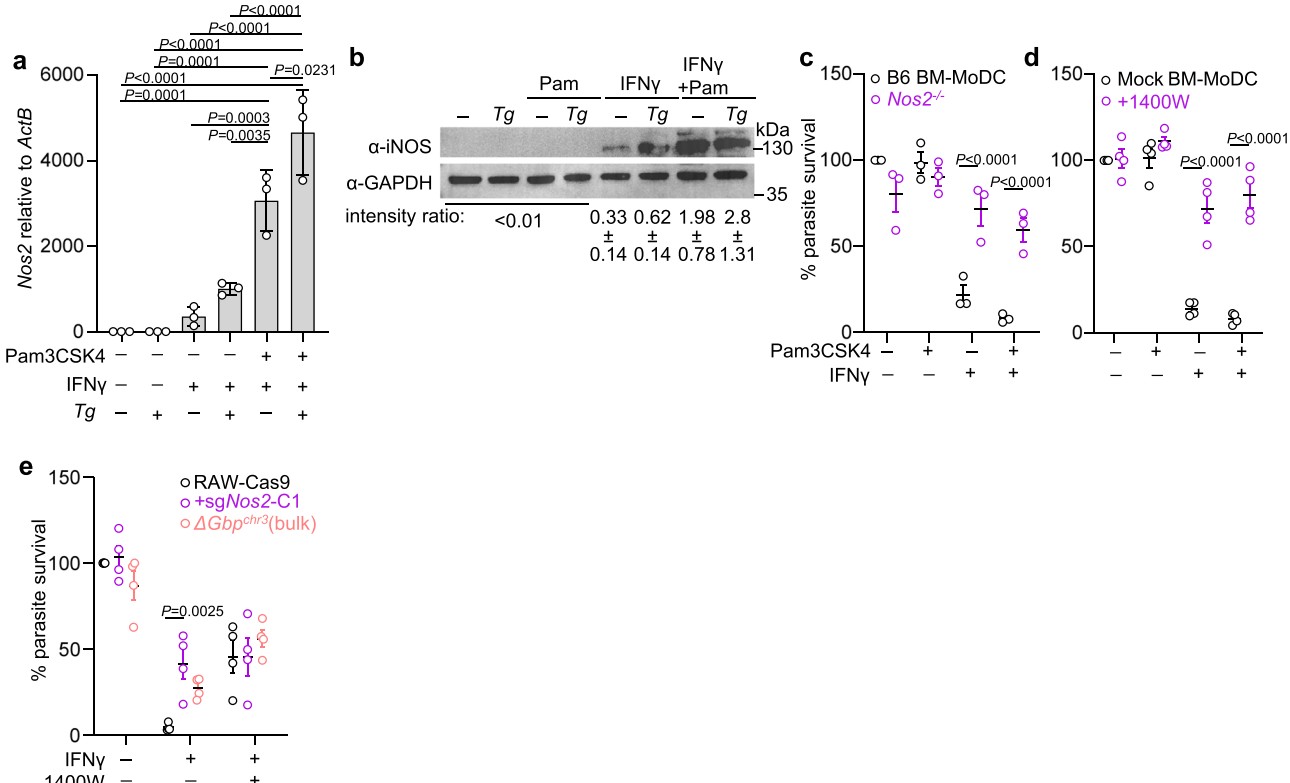

**Fig. 4 | iNOS is upregulated by IFNγ and *T. gondii* infection and necessary for chromosome 3 GBP-mediated *T. gondii* clearance in BM-MoDCs and RAW 264.7 macrophages. a–d** BM-MoDCs were stimulated and infected as described in Fig. 1. **a** *Nos2* transcript expression was determined by quantitative RT-PCR at 2 hpi. **b** iNOS protein levels were detected by Western blot at 14 hpi. **c–d** Parasite survival was measured by luciferase assay 14 hpi in B6 (black) or *Nos2*⁻/⁻ (violet) BM-MoDCs (**c**), treated with the iNOS inhibitor 1400W or vehicle control (**d**). **e** Parental RAW 264.7 macrophages stably expressing Cas9, with a CRISPR-mediated deletion of *Nos2*, or deletion of the chromosome 3 GBP locus *ΔGbpchr3* were treated with media or 10 ng/ml IFNγ for 20 h with 1400W or vehicle. Cells were infected with Me49GFP-Luc at MOI 5 and parasite survival was measured at 14 hpi. Error bars represent Mean ± SEM by one-way ANOVA with Tukey post hoc analysis (**a**), 2-way ANOVA with Šidák (**c**, **d**) or Tuckey (**e**) post hoc analysis. $N = 3$ or 4 independent experiments for panels (**a**, **b**, **c**) and panels (**d**, **e**), respectively.

promote parasite clearance independent of IIGs[53]. However, chloroquine, which buffers lysosome acidification, did not rescue parasite growth in IFNγ-treated conditions (Supplementary Fig. 5d). Together these data support a model where chromosome 3 GBPs are necessary for optimal iNOS-mediated parasite clearance.

### iNOS restricts *T. gondii* via production of nitric oxide and RNS

We next asked which effector functions of iNOS were responsible for *T. gondii* clearance. iNOS converts the substrate ʟ-arginine into nitric oxide (˙NO) and L-citrulline (Fig. 5a)[45]. *T. gondii* is an arginine auxotroph and iNOS activation in macrophages has been proposed to limit *T. gondii* growth by depleting free arginine[54,55]. Supplementing ʟ-arginine did not rescue parasite growth in wildtype BM-MoDCs stimulated with IFNγ+Pam3CSK4 compared to *Nos2*⁻/⁻ (Fig. 5b). While this experiment does not definitively exclude arginine depletion as a means of limiting *T. gondii* viability, it prompted us to explore alternative mechanisms of iNOS function.

˙NO is a membrane-permeable gas that does not react with most biological molecules until converted to reactive nitrogen species (RNS) via reaction with strong radicals[56] (Fig. 5a). RNS, including nitrogen dioxide (˙NO₂) and peroxynitrite (ONOO⁻), regulate a range of biological processes by modifying proteins, unsaturated lipids, and nucleic acids[57,58]. NO was significantly induced by *T. gondii* infection in IFNγ-stimulated BM-MoDCs or by Pam3CSK4 and IFNγ priming (Fig. 5c, gray bars), mirroring the pattern of iNOS protein expression (Fig. 4b). We next treated BM-MoDCs with the ˙NO donor DETA NONOate (Fig. 5d) and found that supplementing exogenous ˙NO sufficient to restore IFNγ-mediated *T. gondii* clearance in *Nos2*⁻/⁻ BM-MoDCs. Notably,

NONOate did not restrict *T. gondii* in unstimulated cells (Fig. 5d) consistent with a model where chromosome 3 GBPs and iNOS cooperate to restrict parasite infection (Fig. 4e). Similarly, expressing human iNOS in RAW cells was not sufficient to restrict *T. gondii* in unstimulated conditions even though 400 ng/mL doxycycline treatment led to nitrite (NO₂⁻) levels equivalent to IFNγ treatment (Supplementary Fig. 6a, b).

We next examined the role of reactive oxygen species (ROS) in *T. gondii* clearance. The antioxidant N-acetylcysteine (Supplementary Fig. 7a, NAC) or the mitochondrial ROS scavenger mitoTEMPO (Supplementary Fig. 7b) failed to rescue parasite growth in B6 BM-MoDCs. Moreover, NADPH oxidases were not transcriptionally upregulated in infected RAW 264.7 cells (Supplementary Fig. 7c) and phagosome oxidase (PHOX) components were not enriched at the vacuole in any IFNγ-stimulated autoSTOMP conditions (Supplementary Data 1). Together, these data are consistent with a primary role for RNS downstream of iNOS in GBP-mediated parasite clearance.

Amino acid nitrosylation (R-NO) and nitration (R-NO₂) are post-translational modifications that regulate protein-protein and protein-membrane interactions, turnover, and signaling transduction, for example, by competing for phosphorylation sites[58,59]. Cysteine nitrosylation, is extremely labile, whereas tyrosine nitration is less frequently observed but stable (Fig. 5a)[57,60]. To determine if the PV is a direct target of RNS, we stained infected macrophages with a nitrotyrosine (NO₂Y)-specific antibody (Fig. 5e–g). Following IFNγ or IFNγ and Pam3CSK4 stimulation robust NO₂Y staining co-localized with *T. gondii* GFP signal (Fig. 5e, f top panel) and, often, the region surrounding the parasite signal (Fig. 5f bottom panel). Tyrosine nitration

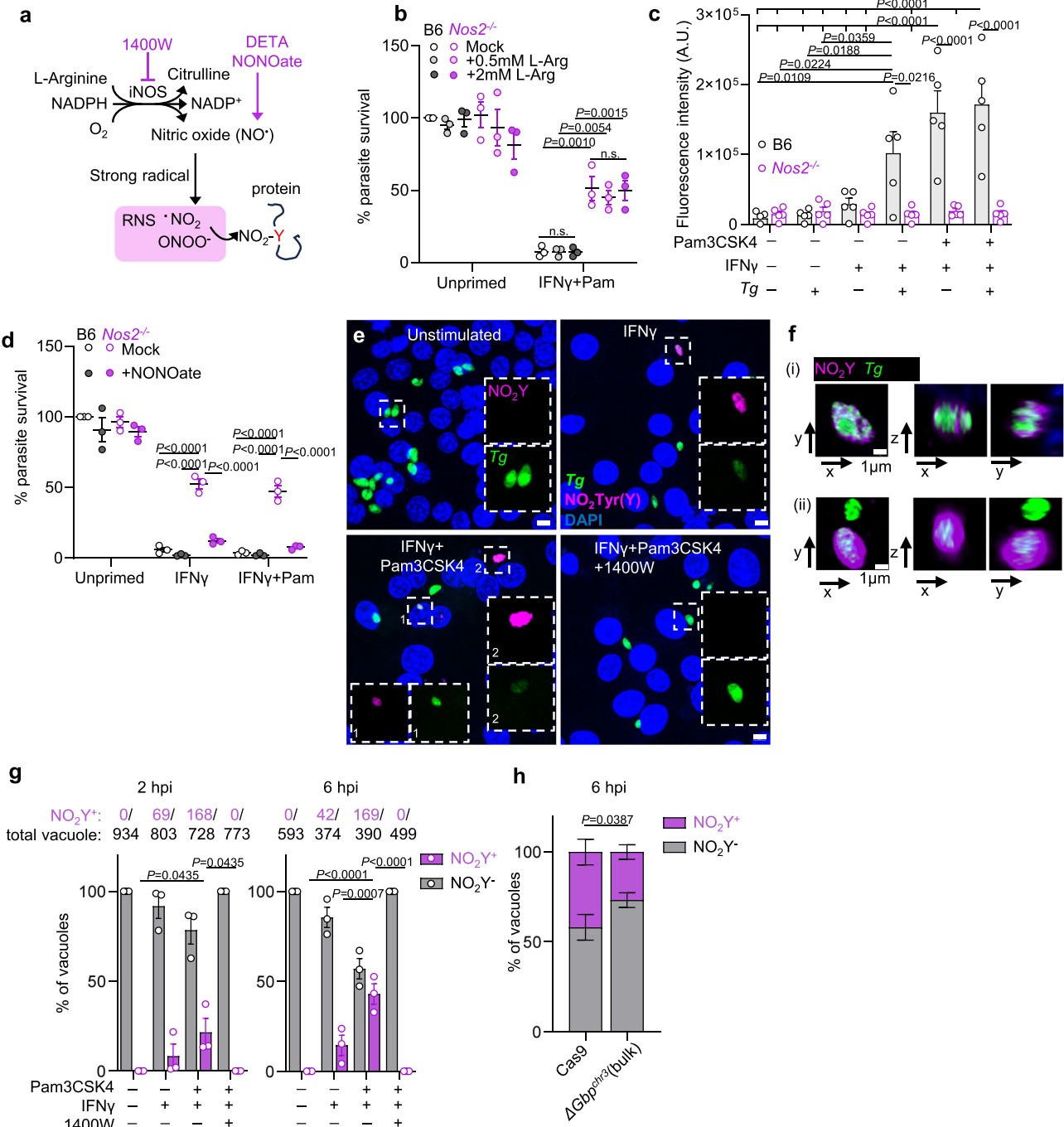

**Fig. 5 | Reactive nitrogen species are necessary for *T. gondii* clearance, leading to nitration of *T. gondii* vacuoles. a** Schematic of ʟ-arginine flux through iNOS, to generate NO synthesis, reactive nitrogen species (RNS). **b–d** BM-MoDCs were stimulated and infected as described in Fig. 1. **b** Parasite growth in normal media or media supplemented with ʟ-arginine one hour before infection was measured by luciferase assay at 14 hpi. **c** NO production was measured by NO-sensitive fluorescent probes at 14 hpi. **d** The NO donor DETA NONOate was added to BM-MoDCs one hour prior to infection, and parasite load was determined by luciferase assay at 14 hpi. **e–h** At 2 and 6 h post-infection, samples were fixed and stained with a nitrotyrosine-specific (NO₂-Y, magenta) antibody to evaluate co-localization with parasite GFP (**e**), high-resolution imaging showing one plane from z-stacks (**f**). Scale bars, 5 μm in (**e**) or 1 μm in (**f**). Co-localization was quantified, in the absence or presence of iNOS inhibitor (1400 W) (**g**), (**h**) the requirement for chromosome 3 GBPs was evaluated in a similar experiment using RAW cells, *N* = 4 independent experiments. **b**, **d**, **g** *N* = 3; (**c**) N = 5 independent experiments. High-resolution Airyscan imaging in (**f**) was done once. Mean ± SEM, two-way ANOVA with Tukey post hoc test. **h** RM two-way ANOVA with Šidák post hoc analysis.

was remarkably selective for the parasite vacuole, as little NO₂Y staining of the host cell was detected. In addition, tyrosine nitration was vacuole selective as macrophages that had been infected multiple times were observed with NO₂Y-positive and NO₂Y-negative vacuoles (Fig. 5f bottom panel). Tyrosine nitration was reversed by iNOS inhibitor treatment (Fig. 5g). Like our parasite clearance result (Fig. 5d),

supplementing ˙NO in unstimulated conditions was not sufficient to induce appreciable vacuole nitration (Supplementary Fig. 7d). This result suggested that IIGs downstream of IFNγ were necessary for vacuole nitration. To test this directly, we evaluated tyrosine nitration in RAW^{ΔGbp-chr3} (Fig. 5h) and found that there was a significant reduction in NO₂Y-positive vacuoles compared to RAW-Cas9 cells. Although it is

formally possible that chromosome 3 GBPs play a signaling role distal to the vacuole, these data are consistent with the model that GBP recruitment to the vacuole is necessary for optimal RNS-mediated clearance of parasites downstream of iNOS activation and synthesis of ˙NO.

### iNOS collapses the IVN, limiting parasite escape from GBP clearance

IIG localization to the PVM has been associated with membrane ruffling, permeabilization to cytosolic fluorescent molecules, and *T. gondii* death[19,20,26,27,30]. To understand how iNOS cooperates with GBPs to mediate parasite clearance, we asked if iNOS functions upstream or downstream of GBP recruitment to the vacuole. RAW-Cas9 cells and RAW[ΔNos2] cells had a similar frequency of GBP2+ vacuoles (Fig. 6a–bi), indicating that iNOS was not necessary for GBP2 recruitment. It was immediately apparent that many of the GBP2-positive vacuoles in RAW[ΔNos2] cells contained morphologically normal, dividing parasites (Fig. 6a, arrowhead). The number of parasites per GBP2-positive vacuole was significantly increased in RAW[ΔNos2] compared to RAW-Cas9 cells (Fig. 6b-ii). However, the number of parasites per GBP2-negative vacuole was similar, indicating that parasites were not simply growing more efficiently in RAW[ΔNos2] than RAW-Cas9 cells (Fig. 6b-ii). These data indicate that IIG recruitment is not sufficient to optimally restrict parasite growth in the absence of iNOS, supporting a model of cooperative clearance downstream of IIG recruitment.

To evaluate the vacuole-autonomous cooperation between these pathways we generated RAW-Cas9 cells expressing tetracycline-inducible GBP2 tagged with blue fluorescent protein on the N terminal (mBFP2-GBP2). These cells were stimulated with IFNγ and Pam3CSK4 with or without iNOS inhibitor, then infected with GFP-expressing *T. gondii* (Fig. 6c, Supplementary Fig. 8). The localization of mBFP2-GBP2 to the vacuole was initiated as early as the first frame (before 1 h) and as late as 16.8 h post-infection (Fig. 6c, left). Loss of parasite-GFP was associated with morphological changes indicative of viability loss (Supplementary Fig. 8a, Supplementary Movie 1). GFP loss on GBP2+ vacuoles were observed from 1 h 20 min to 19 h post-infection (Fig. 6c, Supplementary Fig 8b, c). The iNOS inhibitor 1400W did not change the rate of mBFP2-GBP2 targeting, however, it did delay the median time to loss of GFP signal by 10-h (Fig. 6c, left). Although some parasites were killed when iNOS was inhibited this was significantly reduced compared with IFNγ and Pam3CSK4 treatment (Fig. 6c, right). Confirming static image analysis (Fig. 6a, b) dividing parasites were observed in mBFP2-GBP2+ vacuoles. Two new phenotypes were observed exclusively when iNOS was inhibited. There was a significant increase in GBP-targeted parasite egress from host cells. Moreover, parasites were observed shedding GBP from the vacuole (Fig. 6c, Supplementary Fig. 8, Supplementary Movie 2–4). Cumulatively, these data indicate that iNOS is necessary for the efficient clearance of parasites targeted by GBP2.

To understand how iNOS impacts parasite vacuole structure we evaluated the PV ultrastructure in infected RAW-Cas9 cells using transmission electron microscopy (TEM) (Fig. 6d). Consistent with previous reports[17,20,27,30], IFNγ and Pam3CSK4 treatment led to PVM ruffling, discontinuity, and a significant reduction in the IVN area (Fig. 6d, e). Unexpectedly, blocking iNOS activity rescued the IVN volume (Fig. 6d, e, double arrow, and inset) although morphological changes to the PVM were still observed in vacuoles containing one (Fig. 6d, asterisk) or two parasites (Supplementary Fig. 9a). We next sought to distinguish between the possibilities that RNS was required at the onset of infection, to limit IVN biogenesis or RNS could disrupt the IVN once the infection had been established. RAW[ΔNos2] cells were treated with IFNγ and Pam3CSK4 for 6 h, then pulsed with vehicle or the nitrogen donor NONOate for another 4 h before harvesting for TEM (Supplementary Fig. 9b, c). NONOate treatment led to a significant decrease in the IVN area compared to vehicle, indicating that

RNS can collapse the established IVN. In addition, we confirmed that pulsing NONOate into RAW[ΔNos2] cultures after infection was sufficient to induce parasite clearance by luciferase assay (Supplementary Fig. 9d). Together, these data are consistent with a model where IIGs and RNS target distinct aspects of parasite vacuole biology, moreover, they indicate that PVM disruption by the IIGs alone may not be sufficient for efficient parasite killing.

To understand the requirement of iNOS and the chromosome 3 GBPs on vacuole integrity, we labeled the host cell cytosol with Cell-Mask, blinded the samples, then evaluated the area of the CellMask-negative IVN 'void' surrounding each parasite (Fig. 6f). In RAW-Cas9 cells, IFNγ treatment led to a significant reduction in the void area between the *T. gondii* and host cytosol (Fig. 6g, black). Similar to the TEM results (Fig. 6e), the CellMask-negative void was rescued in RAW[ΔNos2] cells treated with IFNγ or IFNγ and Pam3CSK4 (Fig. 6g, violet). RAW[ΔGbp-chr3] cells also had a significant rescue of the void area (Fig. 6g, red). Together these data are consistent with the conclusion that GBP localization to the vacuole and ruffling of the PVM is necessary but not sufficient for efficient parasite clearance in mouse macrophage. In the absence of iNOS activity, parasites can evade GBP attack via egress, GBP shedding, and replication in GBP-targeted vacuoles. Optimal parasite killing requires transcriptional upregulation of iNOS, synthesis of ˙NO and RNS which modify the PV and collapse the IVN to facilitate parasite clearance in cooperation with the chromosome 3 GBPs (Fig. 6h).

## Discussion

Here we have shown that iNOS expression in murine macrophages is necessary for efficient, vacuole-autonomous parasite clearance by GBPs. Early reports evaluating the role of iNOS in *T. gondii* infection[55] used reversible L-arginine analogs to inhibit iNOS and could not discriminate between L-arginine deprivation and RNS production as the mechanism of parasite control[61]. Our data exclude L-arginine deprivation and implicate ˙NO synthesis as the critical pathway for killing in a manner that depends on the chromosome 3 GBPs. Similar to indolamine dioxygenase, iNOS has been hypothesized to promote parasite 'stasis' in a manner that complements IRGM3 function, however, *T. gondii* growth rate was not directly measured in this study[62]. Our data align with a published result from the Yamamoto lab showing that iNOS inhibitor treatment increased the number of parasites per vacuole in peritoneal macrophages from WT mice relative to chromosome 3 GBP-deficient mice[24]. However, since iNOS inhibition did not change the infection rate, this relationship was not explored further. In addition, skeletal muscle cells treated with IFNγ and TNF have been shown to recruit IRGb6 to the vacuole and produce nitrite in response to Type I and Type II parasite infection; however, a direct relationship between these effector mechanisms was not tested[63]. Perhaps the most critical evidence supporting the importance of iNOS in *T. gondii* clearance is the observation that parasites have evolved strategies to inhibit the ˙NO production[64,65] and promote the proteasomal degradation of iNOS[64]. The Knoll lab found that *T. gondii* patatin-like protein (*Tg*PL1) limited nitrite synthesis in LPS and IFNγ-stimulated macrophages and protected parasites from degradation[66,67], however, the precise mechanism by which *Tg*PL1 interrupts the iNOS/˙NO axis is not clear.

In addition to establishing a requirement for iNOS in IFNγ and GBP-mediated parasite clearance, we have found that the PV is robustly and selectively modified by reactive nitrogen species (Fig. 5e). The high levels of ˙NO flux observed here are expected to lead to dynamic nitration and nitrosylation of lipid, protein and/or nucleic acid substrates[57]. Although nitrated parasite nucleic acids were not detected using 8-nitroguanine-specific antibodies (data not shown), it is likely that other targets of RNS, including reactive cysteines and unsaturated lipids[57,58], also participate in IVN collapse. Unsaturated phospholipid tails in lipid bilayers can serve as nitrogen sinks that

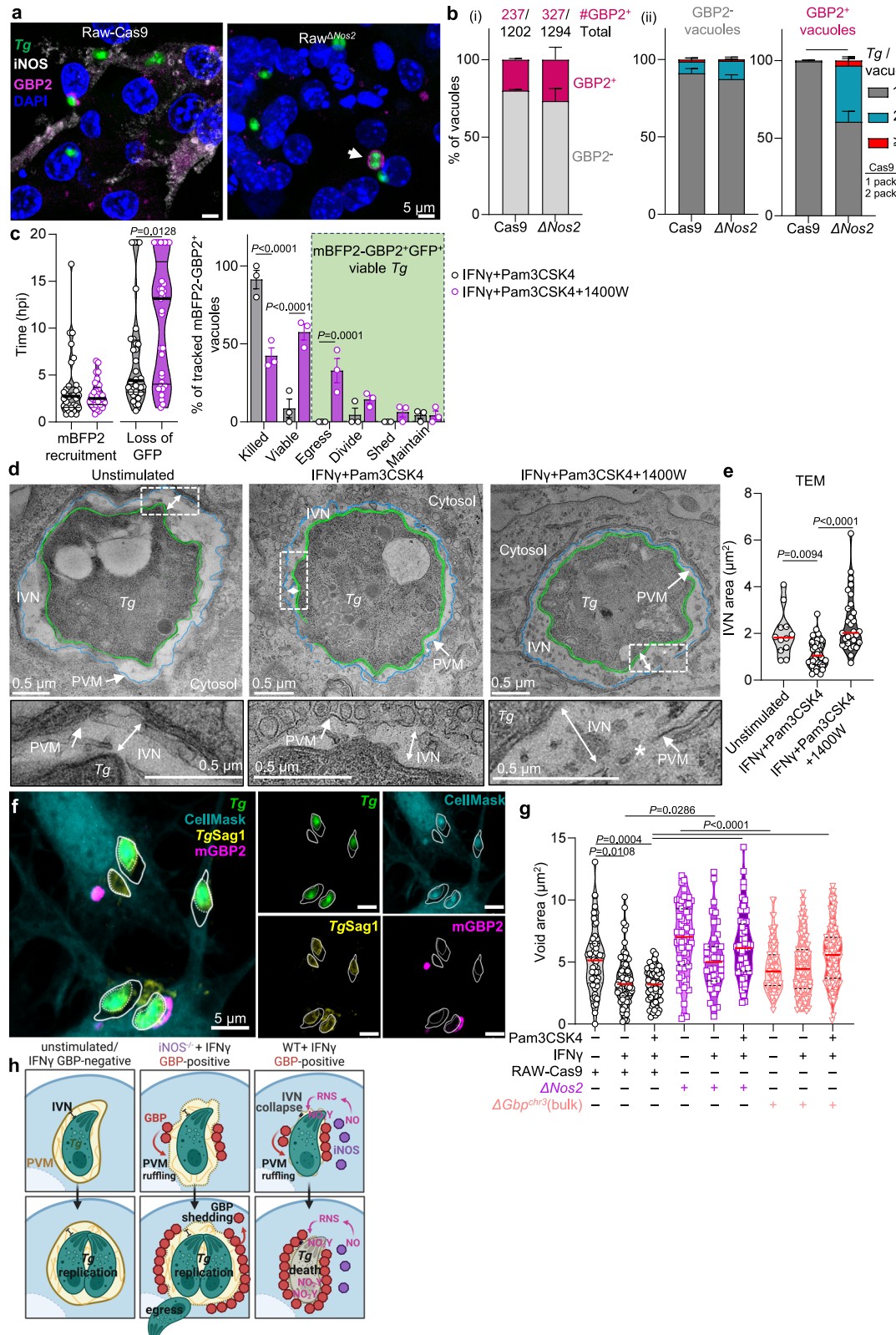

target membrane integral proteins for nitration and nitro-lipidation[68]. However, reliable identification of nitrated lipid species is challenging and there are no tools to evaluate the sub-cellular location of modified lipids. Until recently, tyrosine nitration was posited to be irreversible[57,60], which may be why this modification was readily detected in our study. While it is possible that specific $NO_2Y$-modified substrate(s) regulate IVN collapse and parasite clearance, we favor a model where functional redundancy of nitrated/nitrosylated proteins and or lipid targets mediate this biology on GBP-targeted PV[57]. Future mass spectrometry experiments will be required to identify protein targets of RNS and test the role of these modifications on a case-by-case basis.

The IVN (also called the membranous nanotubular network or, in *Plasmodium* the tubovesicular network) is a series of host-derived lipid

**Fig. 6 | iNOS leads to intravacuolar network collapse and is necessary to restrict parasite growth in GBP-targeted vacuoles. a, b** RAW-Cas9 or RAW$^{\Delta Nos2}$ cells were infected with Me49GFP-Luc as described in Fig. 4. 6 hpi, samples were fixed and stained for GBP2 (magenta) and iNOS (white) (**a**), arrow indicates dividing parasites in GBP2-positive vacuole. **b** The frequency of GBP2-targeting (i), and the number of parasites in GBP2-negative (gray) or GBP2-positive (magenta) vacuoles was quantified (ii). $N = 3$ independent experiments, Mean ± SEM, two-way ANOVA with Šidák post hoc analysis. **c** Time-lapse of RAW-Cas9 cells expressing inducible GBP2 fused to mTagBFP2 (mBFP2-GBP2) primed with IFNγ and Pam3CSK4 alone or with 1400W then infected with Me49GFP-Luc. GFP$^+$ parasites targeted by mBFP2-GBP2 were quantified. Images were taken every 20 min between 1 and 20 hpi. Violin plots showing the minimum, maximum, and median (with quartiles) time to GBP2 targeting and time loss of GFP signal on mBFP2-GBP2$^+$ parasites (left). Bar graphs showing the proportion of non-viable parasites (mBFP2-GBP2$^+$GFP$^-$ and mBFP2-GBP2$^-$GFP$^-$), viable parasites (including mBFP2-GBP2$^+$GFP$^+$, egressed mBFP2-GBP2$^+$GFP$^+$, and mBFP2-GBP2 shedding GFP$^+$ populations) and frequency of mBFP2-GBP2$^+$GFP$^+$ parasites division. IFNγ and Pam3CSK4 $N = 68$ vacuoles; IFNγ and

Pam3CSK4 plus 1400W $N = 67$ vacuoles across three biological replicates. Left, two-tailed Mann–Whitney test; right, two-way ANOVA with Šidák post hoc analysis. **d–e** Vacuole ultrastructure was evaluated by TEM in RAW-Cas9 cells that were unstimulated, treated with IFNγ and Pam3CSK4 with or without 1400 W then infected for 6 h. Insets show the intravacuolar network (IVN) between the parasite plasma membrane (green line, $Tg$) and the parasite vacuole membrane (blue line, PVM), * indicates breaks in the PVM (**c**). **d** The IVN area were quantified. Representative images from one experiment, Kruskal–Wallis test with Dunn's post hoc analysis. **f, g** To measure vacuole integrity, RAW-Cas9 (circle), RAW$^{\Delta Nos2}$ (square) or RAW$^{\Delta Gbp-chr3}$ (triangle) cells were infected with Me49-GFP-Luc as described in Fig. 4. 6 hpi, samples were fixed and stained with CellMask (aqua) and $Tg$SAG1 (yellow). The IVN area between the cell mask signal (solid line) and the parasite GFP and/or SAG1 signal (dotted line) was quantified (**g**). Pooled data of 2 independent experiments showing measurements of individual vacuoles, Kruskal–Wallis test with Dunn's post hoc analysis. **h** Model for cooperative parasite clearance by iNOS and chromosome 3 GBPs.

channels connecting the PVM to the parasites and parasites to each other[3,36]. Initially, the IVN was thought to mediate nutrient acquisition from the host, however, emerging data suggest that the IVN provides structural support for the parasite and is necessary for synchronous division[69,70]. Many parasite rhoptry and dense granule proteins traffic to the IVN although their roles in this compartment are not entirely clear. It is possible that RNS-modification of parasite proteins in the IVN is leading to growth arrest. Although a series of rhoptry proteins were identified in our autoSTOMP dataset, we did not identify IVN-associated dense granule proteins (Supplementary Fig. 1). This may be related to the low coverage of this technique (~1.0% of 8,315 *T. gondii* proteins annotated in UniProt) and/or the low abundance of individual GRA proteins rather than their true absence from the GBP2-targeted region. Future studies using autoSTOMP to target the NO$_2$Y-positive vacuolar regions may identify host and/or parasite proteins targeted by RNS as well as mediators of downstream vacuole clearance.

Discovering this role for iNOS has also revealed that GBP recruitment is necessary but not sufficient for optimal IFNγ-dependent parasite clearance in murine macrophages, as previously thought. Our study is consistent with previous TEM data establishing PVM 'ruffling' by the chromosome 3 GBPs[24] or GBP1[27]. However, our data indicate that these morphological changes to the PVM may not be sufficient for parasite death as iNOS inhibition leads to the enrichment of GBP2 positive vacuoles containing multiple parasites, despite PVM deformity[19,30]. The increased rates of egress in conditions when iNOS is inhibited indicate that parasites may be able to sense and respond to changes in PVM integrity engaged by the GBPs. Thus, cells that can upregulate an RNS partner mechanism may be more efficient at limiting parasite spread. On the other hand redox stress is a significant cost to host cell biology. It remains to be determined whether other cell types have evolved to efficiently kill *T. gondii* with the GBP/IRG system independent of RNS or if eNOS/nNOS, which are expressed at far lower levels than iNOS, can participate in parasite killing. Notably, a '2-hit model' of microbial clearance by the GBPs is conceptually consistent with the mechanism discovered by Gaudet et al. where human GBP1 localization to bacterial membranes must partner with an executioner molecule (apolipoprotein (APO) L3) to mediate *Salmonella* killing[71]. The role of APOL3 and iNOS in *T. gondii* clearance from human macrophages remains to be explored. In comparison to the human system where GBP1 is necessary and sufficient for vacuole disruption[30,72], in mice IFNγ upregulates the expression of many, partially redundant of IIGs. IRGa6[18], IRGb6[17], IRGd[16], GBP1[27], GBP2[28], and GBP7[29] have been shown to be necessary for optimal parasite clearance while others, including GBP3, GBP5, and GBP6, can be recruited to the vacuoles already decorated with GBPs[26]. The precise function of each IIGs on the PVM is not fully understood, and future studies will be necessary to discern which IIGs are necessary for iNOS-mediated IVN collapse.

Finally, our study is the first to formally demonstrate that iNOS expression in myeloid cells is necessary to control *T. gondii* infection in vivo. The lysM promoter drives expression on a range of bone marrow-derived myeloid cell lineages[48]. While iNOS was not robustly detected in peritoneal neutrophils, we cannot exclude the contribution of neutrophils at earlier time points or other tissues. Future studies using more selective cre-drivers (eg. Tie2-cre[73], CX3CR1-cre[74], CD11C-cre[75]) may inform the role of iNOS in specific macrophage, dendritic cell, or neutrophil populations in host resistance to *T. gondii*. Previous studies showed *Nos2*$^{-/-}$ mice succumb to *T. gondii* by 4 weeks post-infection following IP and oral infection[46,47]. Lethality was initially attributed to defective parasite restriction in the brain[47]; however, subsequent studies demonstrated that iNOS is necessary to control hepatic parasite burden[46], and iNOS expression in irradiation-sensitive cells was largely responsible for *T. gondii* protection[76]. Interestingly, *Nos2*$^{-/-}$ animals had fewer necrotic lesions in the liver and intestine than wild-type animals, indicating iNOS-mediated parasite clearance comes at the cost of inflammatory damage-to-self[46]. In our studies, not only did *Nos2*$^{-/-}$ mice fail to control parasite burden in the peritoneum, liver, spleen, and lung, they met euthanasia requirements by 10 days post-infection (Fig. 3), which was more rapid than previous reports. While it is possible that this discrepancy is due to differences in the gene interruption (our study used deletion of *Nos2* exons 12 and 13[77] previous studies used deletion of *Nos2* exons 1–4[46,47]), the 129Sv mice used to generate the knockout alleles are notably more resistant to acute infection than C57BL/6 mice[78]. Thus, the limited number of backcrosses used in previous studies (F2 crosses[47] or 5 backcrosses[46] versus 11 backcrosses[77] to C57BL/6 mice used) likely accounts for the delayed lethality. Taken together, this study establishes a new role for iNOS in vacuole autonomous clearance of *T. gondii* by the chromosome 3 GBPs in murine myeloid cells.

## Methods
### Inclusion and diversity
We support inclusive, diverse, and equitable conduct of research. Citations in this manuscript were selected to reflect scientific contribution as well as inclusive gender and geographical diversity, whenever possible.

### Animals and husbandry
C57BL/6 (Jax #:000664), *Nos2*$^{-/-}$ (B6.129P2-*Nos2*$^{tm1Lau}$/J, Jax #:002609), and CBA/J (Jax #:000656) mice were purchased from the Jackson Laboratory. *LysM$^{cre}$Nos2*$^{fl/fl}$ mice[79] were a gift from Drs. André Marette from Laval University and Dr. Frederick H. Epstein from University of Virginia. Animals were housed in a facility with a 12hr dark/12hr light cycle with temperature of 18–23 °C and 40–60% humidity. Mice were bred and housed in accordance with the University of Virginia

Institutional Animal Care and Use Committee, Association for Assessment and Accreditation of Laboratory Animal Care, and Institutional Animal Care and Use Committee Protocol 4107-12-21.

## Cell culture

Bone marrow was isolated according to previous publications[80,81]. Bone marrow was seeded at $1 \times 10^6$ in RPMI (Gibco,11875119) + 10%FBS containing 20ng/mL of recombinant GM-CSF (PreproTech, 315-03), fresh differentiation media was added on day 3 and BM-MoDCs were plated for infection on day 7. Loosely attached and adherent cells were collected, counted using trypan blue on the Countess cell counter (Thermo Fisher Scientific), and seeded on tissue culture treated 96-well (Corning 3596) or 24-well plates (Fablab, FL7123). RAW264.7 cells were purchased from ATCC (cat. # TIB71) and maintained according to the ATCC protocol. Plated cells were rested at least 12 h before stimulation.

## Parasite infection in vitro

Type II Me49 parasites expressing GFP and luciferase (Me49GLuc) were maintained for up to 10 serial passages on confluent, primary human foreskin fibroblasts (HFFs) in 3 mL of DMEM (Thermo Fisher Scientific, 11965118) + 10% FBS. Intracellular parasites were released by scraping and passage through 22g blunt end needles (Instech Laboratories, LS22/6S). The parasite suspension was centrifuged at 40 × $g$ for 3 min to remove large debris and then washed twice by centrifuging at 300 × $g$ for 6 min each. The pellet was resuspended in DMEM + 10%FBS and counted on the hemocytometer. Parasites were then diluted to the indicated multiplicity of infection (MOI) so that 5 μL of suspension was spiked in every well for infection.

For BM-MoDC infections, cells were seeded at $1 \times 10^6$ cell/mL, and media was changed to OptiMEM (Gibco, 31985070) + 1% FBS containing 70 ng/mL of mIFNγ (R&D, 485MI100CF) 20 h before infection. Three hours before infection, Pam3CSK4 (Invivogen, tlrl-pms) was added to the wells at a final concentration of 100 ng/mL. Cells were then infected with Me49GLuc at MOI 1.5.

For infection with RAW 264.7 cells, cells were seeded at $1.5 \times 10^5$ cell/mL in DMEM + 10%FBS and media was changed to DMEM + 10% FBS containing 10 ng/mL of mIFNγ 24 h before infection and at the last 3 h, Pam3CSK4 was spiked in to a final of 100 ng/mL. Cells were then infected with Me49GLuc at MOI 5.

At 14 h post-infection, parasite burden was measured using Steady-Luc Firefly HTS Assay Kit (Biotium, 30028-L2) according to the manufacturer's instructions. The parasite burden was calculated using relative light units (RLU) read from Cytation 5 plate reader (BioTek).

The following inhibitors were added 1 h before infection, unless otherwise noted in the figure legends: 1400W-HCl (100μM, Selleck Chemicals, S8337), L-Arginine (0.5 or 2mM, MP Biomedicals, 0219462625), DETA NONOate (300μM, Cayman Chemical Company, 82120), N-aceyl-L-Cysteine amide (1mM, Cayman Chemical Company, 25866), mitoTEMPO (200 or 500μM, Sigma, SML0737-5MG). Chloroquine (100μM, Thermo Fisher Scientific, 455240250) was added 1 h after infection to avoid impacting parasite invasion.

## Immunofluorescence staining and imaging

BM-MoDCs or RAW 264.7 cells were seeded on poly-d-lysine (MP Biomedicals, 0215017550) coated coverslips (Harvard Apparatus, 64-0712) in 24-well plates stimulated and infected as described above. At the time of collection, media was aspirated and cells were fixed by 4% PFA (Electron Microscopy Sciences, 15710) for 15 min at room temperature. Samples were washed twice with PBS, permeabilized with PBS + 1% Triton X-100 (Fisher) for 30 min, and blocked with PBS + 5% BSA (Fisher) or serum of the host of secondary antibodies (Jackson ImmunoResearch). Coverslips were incubated with primary antibodies overnight at 4 °C, followed by 3 washes with PBS, and incubation with fluorescent secondary antibodies. After 3 washes and DAPI staining

(Thermo Fisher Scientific, D1306), coverslips were mounted in the ProLong Gold Antifade Mountant (Thermo Fisher Scientific, P36930) or Vectashield Mounting Medium (Vector Laboratories, H-1000-10). Coverslips were imaged on Zeiss LSM 880 (Carl Zeiss) using 40x (Plan-Apochromat NA1.3, Oil DIC M27) or 63x (Plan-Apochromat NA1.4, Oil DIC M27) lenses.

Primary antibodies used: α-GBP2 (Proteintech, 11854-1-AP, 1:500 dilution), α-*Toxoplasma* polyclonal (Thermo Fisher Scientific, PA1-7253, 1:500 dilution), α-*Tg*SAG1 (Thermo Fisher Scientific, MA518268, 1:100 dilution), α-iNOS (BD Biosciences, 610328, 1:500 dilution),, α-nitrotyrosine (EMD Millipore, 06–284, 1:200 dilution). Following secondary antibodies were used at 1:500 dilution: Goat anti-Rabbit IgG (H + L) Highly Cross-Adsorbed Secondary Antibody, Alexa Fluor 594 (Thermo Fisher Scientific, A11037); AffiniPure Donkey Anti Mouse IgG (H + L), Alexa Fluor 594 (Jackson ImmunoResearch, 715-585-150); Donkey anti-Rabbit IgG (H + L) Highly Cross Adsorbed Secondary Antibody, Alexa Fluor 647 (Thermo Fisher Scientific, A31573).

## autoSTOMP procedure and LC-MS

AutoSTOMP was performed as published previously[33,34]. Briefly, B6 BM-MoDCs were seeded on 18 mm coverslips (Thomas Scientific, 1217N81) at 1.35 x 10$^6$ cells per coverslip in 12 well plates. Cells were treated with RPMI + 10%FBS containing 70 ng/mL of mIFNγ 16 h before the infection and at the last 3 h, Pam3CSK4 was added to a final concentration of 100 ng/mL. Cells were then infected with Me49GLuc at MOI 1.5. Samples were harvested at 2 h.p.i by fixing in cold absolute methanol (Fisher) for 20 min on ice and stained following the IF staining protocol including avidin/biotin blocking (Vector Laboratories, SP-2001) following the manufacturer's instructions. Slides were stored at −30 and mounted immediately prior to autoSTOMP imaging in 1 mM biotin-dPEG3-benzophenone (Quanta BioDesign, Biotin-BP) in 50:50 (v/v) dimethyl sulfoxide (DMSO)/water. Slides were subjected to autoSTOMP procedure on Zeiss LSM 880 microscope (Carl Zeiss) with a Chameleon multiphoton light source (Coherent) and a 25x oil immersion lens (LD LCI Plan-Apochromat 25×/0.81 mm Korr DIC M27). Regions of interest were defined as the 1 pixel region surrounding all *T. gondii* signal was defined as the 'PVM' MAP or the region of GBP2+ staining overlapping or immediately adjacent to the *T. gondii* signal was defined as the 'GBP2' MAP. The specificity of photo-biotinylation was confirmed by Streptavidin-594 secondary staining and 'dark controls,' where samples were not exposed to 2 photon excitation, were performed on unstimulated (N = 2, labeled 'Dark 1' and 'Dark 2') and IFNy stimulated (N = 1, 'Dark 3') conditions as described in ref. 33. The dilate control (n = 3) was performed on the IFNγ-stimulated conditions where the PV region was dilated by 1 pixel to biotinylated the region surrounding but excluding the region targeted for the 'PVM" sample. A detailed protocol and source codes are available at [https://github.com/boris2008/Sikulix-automates-a-workflow-performed-in-multiple-software-platforms-in-Windows].

After autoSTOMP crosslinking samples were washed, dissociated from the coverslips in 8 M urea lysis buffer (100 mM NaCl, 25 mM Tris, 2% SDS, 0.1% tween 20, 2 mM EDTA, 0.2 mM PMSF, and 1x Roche cOmpleteProtease inhibitor). Lysates were treated with benzonase (Sigma, E1014-25KU) and RNase (Sigma, 10109142001) to reduce viscosity then subjected to affinity purification using Pierce Streptavidin Magnetic Beads (Thermo, 88817). Enriched proteins were eluted by boiling at 96 °C for 5 min in Laemmli SDS buffer (10μM DTT, 0.0005% Bromophenol blue, 10% Glycerol, 2% SDS, 63 mM Tris-HCl pH 6.8) and ran on a pre-cast Novex Tris-Glycine Mini Protein gel (Thermo Fisher Scientific, XP04125BOX) at 70 V for 12 min. 1 cm gel fragments were cut from each lane and submitted to University of Virginia Biomolecular Analysis Facility for mass spectroscopy analysis. The samples were run on a Thermo Orbitrap Exploris 480 mass spectrometer system with an Easy Spray ion source connected to a Thermo 75 μm × 15 cm C18 Easy Spray column (trap column first). Mass spectra data

were analyzed using MaxQuant (versions 1.6.15.0) following the published pipeline and maxLFQ data were analyzed in Perseus (version 2.0.3.1) using Student's t-test (permutation-based FDR)[33,34]. Plots were generated using R (version 4.2.0) and GraphPad Prism 9.

## Mouse infection

Age and sex-matched animals were infected between 12–18 weeks old. 2 weeks before infection, dirty bedding was mixed to normalize the microbiota. Me49GLuc cysts were passed in vivo ib CBA/J mice to harvest cysts from brain tissue, as previously published[82]. The brain lysate was 1:10 diluted in the PBS and stained with rhodamine-labeled dolichos biflorus agglutinin (DBA-red, Vector Laboratories, RL-1032-2). GFP and DBA-red double-positive cysts were counted and diluted to 5 cysts per 200 μL of PBS. Mice were injected with 200 μL of cyst solution and were monitored using a humane endpoint scoring system based on weight loss, posture, appearance, and activity. Moribund mice were euthanized according to the ACUC protocol.

## Flow cytometry

Cells in the peritoneal cavity were isolated by lavage with 10 mL of cold PBS. Cell number was determined by Countess hemocytometry (Thermo Fisher Scientific). $1.2 \times 10^6$ total cells were isolated for FACS staining in a round bottom 96-well plate. Cells were pelleted down at 1,500 x g at 4 °C for 5 min and cell pellets were resuspended in 100 μL FACS buffer (PBS + 4% FBS + 0.5 mM EDTA) containing Fc block (BioLegend, 101302, 1:200). After 15 min incubation at 4 °C, surface markers were stained with following antibodies at 1:200 dilution at 4 °C for 30 min: CD11b-Pacific Blue (BioLegend, 101224), CD45-BV650 (BioLegend, 103151), CD11c-BV711 (BioLegend, 117349), Ly6C-PerCP/Cy5.5 (BioLegend, 128012), F4/80-PE/Cy7 (BioLegend, 123113), I-A/I-E-AF647 (BioLegend, 107618), Ly6G-APC/Cy7 (BioLegend, 127624). Staining was stopped by 2 washes in FACS buffer, fixation, and permeabilized using BD Cytofix/Cytoperm fixation/permeabilization kit (BD Biosciences, 554714). Intracellular iNOS was stained with iNOS-AF594 (BioLegend, 696803, 1:200) in the FACS buffer for 1 h at room temperature in the dark. Cells were then washed twice using BD Perm/Wash buffer and resuspended in 200 μL of FACS buffer. Single stains were prepared using UltraCom eBeads (Thermo Fisher Scientific, 01-2222-42) as well as fluorescence minus one (FMO) controls. Samples were run on an Attune Flow Cytometer (Thermo Fisher Scientific) equipped with 405 nm, 488 nm, 561 nm, 637 nm lasers, and 14 detector channels. Data were analyzed using FlowJo (v10.8) and gated using FMOs.

## Genomic DNA preparation from tissue

Tissues were dissected and flash-frozen on liquid nitrogen. 1 μL of UltraPure water (Thermo Fisher Scientific, 10977015) was added per 1 mg of tissue, then bead beaten (Qiagen, 69989) using a Qiagen Tissuelyzer (Qiagen) for 3 min at 25 Hz. 20 μL of tissue lysates were taken for DNA isolation using DNeasy Blood & Tissue Kit (Qiagen, 69506) according to the manufacturer's instruction. Eluted DNA was quantified on the NanoDrop One (Thermo Fisher Scientific) and diluted to the same concentration. 50 ng of total DNA from each sample was used for quantitative PCR (qPCR).

## Quantitative PCR

*T. gondii* burden in the tissue was measured by qPCR of *T. gondii* 529-bp repeat element (RE) compared with mouse β-actin as described previously[83] using Taqman probes (Thermo Fisher Scientific). Water or uninfected animals were used as a negative control. Data were analyzed using the ΔCt method. The following TaqMan primer/probes were used: 529-bp RE forward, 5′-CACAGAAGGGACAGAAGTCGAA-3′ and reverse, 5′-CAGTCCTGATATCTCTCCTCCAAGA-3′; probe: 5′-CTA-CAGACGCGATGCC-3 (Integrated DNA Technologies); mouse β-actin: Mm02619580_g (Thermo Fisher Scientific).

For tissue culture experiments, total RNAs were isolated using Quick-RNA Miniprep Kit (Zymo Research, R1055). RNA concentration and quality were assessed by NanoDrop (Thermo Fisher Scientific). 40 ng of total RNA was used for cDNA synthesis using Accuris qMax cDNA Synthesis Kit (Accuris Instruments, PR2100-C-250). Gene expression levels were measured by reverse transcription quantitative PCR (RT-PCR) using PowerUp SYBR Green PCR Master Mix (Thermo Fisher Scientific, A25742). Primers are listed in Supplementary Data 3 and ΔCt was calculated relative to *Actb* gene while relative gene expression levels were quantified using ΔΔCt method normalizing to unstimulated groups. No RT control was included as a negative control. All of the above procedures were carried out according to the manufacturer's instructions.

## Western blot

At the time of collection, media was removed, and cell lysates were prepared by directly adding Laemmli SDS buffer into the well. Protein lysates were transferred out of the plate, sonicated twice on ice, and denatured at 95 °C for 5 min. Lysates were loaded on 10% home-made SDS-PAGE gel and transferred onto a 0.45 μm PVDF membrane (Millipore, IPFL00010) using TransBlot Turbo System (Bio-Rad). The membranes were blocked in TBST containing 5% skim milk at room temperature for 1 h, followed by primary antibody incubation at 4 °C overnight. After three TBST washes, membranes were incubated in TBST containing HRP-conjugated secondary antibodies at room temperature for 1 h. Luminata Forte Western HRP substrate (Millipore, WBLUF0100) and ChemiDoc Imager (Bio-Rad) were used to develop the membrane and acquire images respectively. Volumetric intensities were quantified using Image Lab software (Bio-Rad, version 6.0.0). Primary antibodies used: α-GBP2 (Proteintech, 11854-1-AP, 1:1000 dilution), α-iNOS (BD Biosciences, 610328, 1:1000 dilution), α-GAPDH (CST, 5174, 1:1000 dilution). Secondary antibodies used: Peroxidase AffiniPure Goat Anti-Mouse IgG (Jackson ImmunoResearch, 115-035-003, 1:10,000 dilution), Peroxidase AffiniPure Donkey Anti-Rabbit IgG (Jackson ImmunoResearch, 711-035-152, 1:10,000 dilution).

## NO and nitrite measurement

Intracellular NO levels were measured using OxiSelect Intracellular Nitric Oxide (NO) Assay Kit (Cell Biolabs, STA-800-5). Nitrite levels in the cleared supernatant were measured using Griess Reagent Kit (Thermo Fisher Scientific, G7921) according to the manufacturer's instructions and detected on a Cytation5 plate reader.

## Plasmids

pLIX-hNos2 (Addgene, 110800), psPAX2 (Addgene, 12260), pMD2.G (Addgene, 12259), LentiGude Puro (Addgene, 52963) were purchased from Addgene. CAS9 Blasticidin Lenti plasmid was purchased from Sigma (CAS9BST). Plasmids were expanded in house and isolated using Plasmid Plus Midi Kit (Qiagen, 12945) according to the manufacturer's instructions.

sgRNA targeting the respective gene was designed on E-CRISP and lentivectors expressing sgRNAs were built according to the published protocol[84]. Primers used to generate the sgRNA plasmids were listed in Supplementary Data 3.

LentiGuide-Gbp-Chr3-sg3+sg4 carrying both Gbp-Chr3 sgRNA3 and sgRNA4 generated by first building individual LentiGuide plasmids as stated above. Amplicon containing U6 promoter and sgRNA4 was PCR amplified using Phusion High-Fidelity DNA Polymerase (New England Biolabs, M0530) and assembled into SalI linearized LentiGuide-GBP-Chr3 sgRNA3 plasmid using HiFi DNA Assembly Kit (New England Biolabs, E5520). Primers used were listed in Supplementary Data 3.

pLIX-Gbp-Chr3-repair plasmid containing a neomycin resistance cassette (*NeoR*) targeting chromosome 3 GBP locus was generated from genomic DNA by amplifying 5′ and 3′ homology arms using

primers listed in Supplementary Data 3. The protospacer adjacent motif (PAM) loci of sgRNA were mutated in the amplicons of homology arms. The *NeoR* gene fragment was synthesized by GeneWiz. All fragments were assembled into a pLIX plasmid linearized by NheI and SalI (New England Biolabs) using HiFi DNA Assembly Kit (New England Biolabs, E5520). All plasmids were Sanger sequenced to ensure correct inserts (Eurofins Scientific).

Tetracycline inducible pLV-Hygro-TRE3G-mTagBFP2-GBP2 plasmid was custom made by VectorBuilder and the vector ID is VB230707-1304hwk, which can be used to retrieve detailed information about the vector on vectorbuilder.com. The full plasmid sequence is included in the Supplementary Data 3.

### Lentiviral transduction

HEK-293T cells were seeded 1 day before transfection and transfected with psPAX2, pMD2.G packaging plasmid, and lentivectors using Fugene 6 (Promega, E2691). 24 h after transfection, media was changed. After 48 h viral supernatants were collected, cleared of cell debris using 0.45 μm filter (Celltreat, 229766) and added to target cells in the presence of polybrene (EMD Millipore, TR-1003-G). After overnight incubation, fresh media was added. Two days later, cells were selected in 6 μg/mL of puromycin (Sigma, P8833) or 7 μg/mL blasticidin (Research Product International, B122000) for 2 days. Surviving cells were expanded and single-cell cloned.

### Generation of RAW$^{\Delta Gbp\text{-}chr3}$

Fourteen micrograms of pLIX-Gbp-Chr3-repair plasmid were linearized using NheI and SalI. Purified templates together with 2 μg of LentiGuide-Gbp-Chr3-sg3+sg4 were electroporated into RAW-Cas9 cells using Lonza SF cell line X kit (Lonza, V4XC-2012) on a Lonza 4D-Nucleofector unit (Lonza). Code DS-136 was used for electroporation. Cells were flushed out using warm complete DMEM + 10%FBS and rested for 4 days. 1 mg/mL of G418 (Thermo Fisher Scientific, BP6735) was added for 4 days. G418 was added every 2 passages to keep the selection pressure. Successful insertion of the repair template was validated by PCR using primers listed in Supplementary Data 3.

### CellMask staining

Cells and coverslips were prepared as described above. Fixation is followed by a gentle permeabilization with 0.2% (w/v) BSA and 0.02% (w/v) Saponin in PBS (w/o Mg²⁺, Ca²⁺) for 30 min at RT and then blocked in PBS + 5% BSA for 15 min at RT. Primary antibodies were applied for 1h in the permeabilization buffer as described elsewhere. After three 5 min washes, secondary antibodies were prepared in permeabilization buffer containing 2 μg/ml HCS CellMask Blue Stain (Invitrogen Molecular, H32729) and incubated at RT for 30 min protected from light. Following 4 washes, coverslips were mounted with ~25 μl ProLong Glass Antifade Mountant (Thermo, P36980) and left to fully cure overnight in the dark at RT before imaging or longer-term cold storage. The area of the cell mask negative void and the parasite GFP signal was analyzed in Fiji.

### Transmission electron microscopy

$5.0 \times 10^6$ RAW 264.7 cells were plated into three 10cm dishes and incubated overnight at 37 °C with 5% CO2. Subsequently, appropriate dishes were primed with 10ng/ml mIFNγ and Pam3CSK4 for 24 h prior to infection. An hour before infection, one primed dish was treated with 100 μM 1400W. Both dishes were infected for 6 h with Type II Me49-GFP-Luc at MOI 2, following which cells were washed thrice with ice-cold PBS. In a second experiment, RAW$^{\Delta Nos2}$ cells were seeded, stimulated and infected as described above. 6 h post-infection one dish was treated with DETA NONOate (300 μM, Cayman Chemical Company, 82120) and a second was treated with vehicle control and samples were harvested 4 h later.

Simultaneously, a fixation buffer was freshly prepared with 2.5% glutaraledehyde, 0.1M sodium cacodylate (both Sigma-Aldrich, G5882-50ML, and C4945-10G, respectively), and sterile deionized H₂O. The buffer was applied to the cells, which were then allowed to fix at room temperature for 1 h. Cells were scraped gently into 'sheets' and collected into 2-ml Eppendorf tubes, centrifuged, and washed thrice with 0.1M sodium cacodylate. They were stored at 4 °C in a 0.1M sodium cacodylate buffer with minimal air exposure.

For ultrastructural analysis, cells were washed with 0.1M sodium cacodylate twice, immersed in 1% osmium tetraoxide (Sigma-Aldrich, 75632) in 0.1M sodium cacodylate at RT for 1 h, then washed again. Dehydration was carried out using an ethanol gradient (50%, 70%, 95%, and 100%), each for 10 min. A series of infiltration steps using an increasing ratio of epoxy resin was executed. Polymerization was performed overnight at 60 °C. Ultrathin sections (80 nm) were cut using a Leica EM UC7 Ultramicrotome, collected on 200 mesh copper grids, counterstained with uranyl acetate and lead citrate, and carbon-coated for TEM stability. Samples were stored at RT until imaging. Samples were imaged on a FEI Tecnai F20 equipped with a 4K camera (TSS microscopy). All images were blinded and the area of the intravacuolar network (IVN) space was analyzed in Fiji.

### Live imaging

To visualize RAW-Cas9 cells transduced with tetracyclin-inducible GBP2 tagged with blue fluorescent protein, mTagBFP2, in the N-terminal (mBFP2-GBP2) were plated on poly-D-lysine coated chambered slides (MP Biomedicals, 0215017550) at $2 \times 10^5$/well in Fluoro-Brite DMEM (Thermo Fisher, A1896701) and 10% FBS. Cells were treated with 400ng/ml anhydrotetracycline (400ng/mL, Cayman Chemical Company, cat. 10009542) 44 h before infection. 24 h before infection cells were primed with 10ng/mL mIFNγ, 3 h before infection 10ng/mL Pam3CSK4 was added. Cells were infected with Me49-GFP-luc at MOI = 5. Live imaging was performed at the Keck Molecular Imaging Core Zeiss LSM980 using a 20x objective (Plan-Apochromat, NA 0.8 M27). 3 z-stacks covering 8 microns were acquired at 20-min intervals across a 3x2 tiles scan. Each experiment lasted 20 h. Data are quantified from 1 to 20 h post-infection in Zen Black.

### Antibodies

All antibodies used in this study are listed in Supplementary Table 1

### Statistical analysis

Data were plotted and analyzed in GraphPad Prism 9 and R (version 4.2.0) unless noted otherwise. A normality test was carried out whenever possible to choose appropriate statistical analysis.

### Reporting summary

Further information on research design is available in the Nature Portfolio Reporting Summary linked to this article.

## Data availability

The raw mass spectrometry data of autoSTOMP have been deposited in PRIDE database [https://www.ebi.ac.uk/pride/] under accession code PXD027716. Source data are provided with this paper through Figshare and can be accessed through this link [https://doi.org/10.6084/m9.figshare.24907851]. Source data are provided with this paper.

## Code availability

Source codes are freely available at [https://github.com/boris2008/Sikulix-automates-a-workflow-performed-in-multiple-software-platforms-in-Windows] The source code is also deposited to Zenodo with [https://doi.org/10.5281/zenodo.10674574].

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

## Acknowledgements

The authors want to thank the members of Ewald lab for their feedback. We thank Dr. Hervé Agaisse for his insights and suggestions for this manuscript. We thank Dr. André Marette for sharing the *Nos2^{fl/fl}* mice and for constructive discussion of their use to evaluate iNOS biology in vivo. We also thank Dr. Nicholas E. Sherman and the Biomolecular Analysis Facility Core for LC-MS. This work used TEM sample preparation service in the Advanced Microscopy Facility (Research Resource Identifiers (RRID): SCR_018736). Transmission electron micrographs were recorded at the University of Virginia Molecular Electron Microscopy Core facility (RRID:SCR_019031), which is built with NIH grant G20-RR31199. BAF, AMF, and MEMC are supported by the University of Virginia School of Medicine. This work used Zeiss 980 Confocal system (funded by NIH-OD 025156) in the W.M. Keck Center for Cellular Imaging. M.M.M. is supported by Wagner Fellowship by University of Virgnia. This work is supported by NIGMS R35GM138381 (S.E.E.) and NIAID R21AI156153 (S.E.E.).

## Author contributions

Conceptualization: X.Y.Z., S.L.L., J.U.A., S.E.E.; Research Design: X.Y.Z., S.E.E.; Investigation: X.Y.Z., S.L.L., J.U.A., M.M.M., B.Y., N.K.H.; Data analysis: X.Y.Z., S.L.L.; Image quantification: X.Y.Z., S.L.L., J.S., I.G.B; Manuscript Drafting and Editing: X.Y.Z., S.E.E.; Supervision: S.R., S.E.E.

## Competing interests

The authors declare no competing interests.
