## [Peer Review File · Nature Communications]

iNOS is necessary for GBP-mediated *T. gondii* clearance in murine macrophages via vacuole nitration and intravacuolar network collapseREVIEWER COMMENTS

Reviewer #1 (Remarks to the Author):

I am a little concerned that the PV might be too small for the STOMP technique. The literature values for PV appear to be 1-3 microns, that is at the functional cut-off of the STOMP technique. The authors can conduct control experiments to show that they have resolution in the single micron range. The original STOMP technique utilized a dark control, which was treated the same as the sample except the photo-irradiation was omitted. Such control experiments are necessary to convince the reader that specific proteins from the desired subcellular structure is being selected. The authors can also show that their targeting is accurate by providing images before and after photo-irradiation showing that only the PV structure was targeted. Finally a control experiment showing that STOMP analysis of a different part of the cell (e.g. nucleus or cytoplasm) produces a different set of proteins would also provide necessary reassurance to the reader. Because this submission is only the third instance of using this technique, it requires greater control than established techniques. If the control experiments show positive results the manuscript can be considered further.

The link to GitHub in their methods section was not working. They also need to reference the original STOMP technique: Hadley et al., Determining composition of micron-scale protein deposits in neurodegenerative disease by spatially targeted optical microproteomics. *Elife*. 2015 Sep 29;4:e09579. doi: 10.7554/eLife.09579. PMID: 26418743; PMCID: PMC4630677.

Reviewer #2 (Remarks to the Author):

Summary: In the manuscript titled "iNOS is necessary for GBP-mediated T. gondii restriction in murine macrophages via vacuole nitration and intravacuolar network collapse," Zhao et al. use mice, murine-derived immune cells, and Type II T. gondii parasites to ask how GBPs and iNOS interact to restrict parasite growth. The manuscript includes strong evidence for a collaborative role of GBPs and iNOS to restrict T. gondii in vitro. Authors also provide new evidence that iNOS in LysM-positive myeloid cells influences T. gondii infection in vivo. Nitration as a mechanism for iNOS/NO-mediated parasite restriction is interesting. Collectively, iNOS-related experiments add depth to our knowledge of iNOS in T. gondii infection. Enriching for GBP-positive vacuoles with autoSTOMP was a clever and novel way to leverage heterogeneity in the host response to T. gondii. It revealed new relationships in a complex immune signaling pathway. Comments are largely to expand on logic and interpretation, no additional experiments required.

Comments:

- for non-immunologists, a schematic in Figure 1 outlining the signaling pathways discussed (GBPs, IFN γ , TLRs, iNOS, STATs) will get readers up to speed more quickly
- authors should state why they select GBP2 for autoSTOMP but chromosome 3 GBPs for knockout experiments. is the choice for technical or scientific reasons or both?
- line 251: there is no figure 5i
- line 261-263: because chromosome 3 GBPs are absent in Δ Gbp-chr3 cells (not just absent from PVM), it's possible they play additional roles: signaling, indirect effects on other GBPs, etc.
- line 282: should be 'transmission' not 'transmitted'
- line 311, arginine discussion: evidence for nitration is good. Evidence against arginine depletion is ok (2mM arginine should guarantee parasites are replete with arginine), but not perfect. Arginine dynamics are controlled by multiple host and parasite transporters, enzymes like arginase that can consume huge quantities of arginine, and restricted exchange of subcellular arginine pools. The claim would be strengthened by definitively measuring intracellular arginine or even intraparasitic arginine, but for this paper that is probably excessive. Rather, recommend noting that arginine supplementation experiments do not definitively exclude arginine depletion.
- if authors will pursue myeloid NOS2 further, comparison of LysM-driven Cre with Tie2-driven Cre may be warranted. Tie2 drives deletion in a more complete subset of myeloid cells (with the caveat that Tie2 also drives deletion in the endothelium)

Reviewer #3 (Remarks to the Author):

The intracellular mechanisms activated downstream of IFN-gamma signaling that mediate the destructive (toxoplasma-cidal) and growth restrictive (toxoplasma-static) activities of macrophages and non-phagocytic cells have been the subject of several decades of research. A consensus view has been that certain effectors of cell autonomous immunity are growth restrictive (iNOS and IDO for example) and others are direct cidal (GBPs and p47 GTPases) and that each effector arm can operate independently (for example PMID 19265156, not cited in References). In this manuscript, the authors propose that iNOS is essential for GBP's anti-toxoplasmic effects through nitration of the parasite vacuole and by inducing "collapse" of the intravacuolar network formed normally by invaded parasites. If fully substantiated, the work would represent a significant advance in the field. However, multiple issues in the experimental design and data presentation need to be addressed further.

1. In this paper, a clear definition and distinction should be made between cidal and static effects/endpoints of the activated host cells. Often in the manuscript, the terms "restriction" and "clearance" are used interchangeably. This is evident in the abstract and elsewhere in the manuscript. In Figure 1 the time course is spread over a long duration of time, but the cidal activities are mostly completed within the first 4 hours of the time course as reflected by the drastic attrition of the number of vacuoles present at 14 hpi. In anticipation of the focus on GBPs and iNOS, the expression levels of these IFN-gamma induced effectors should be shown under all four conditions studies: unprimed, Pam3CSK4, IFN-gamma and combination treatments.
2. The statement (heading) used in the results section to describe the autoSTOMP data (Figure 2 and extended data Fig. 2) can be readily misinterpreted and thus can be misleading. It states that "iNOS is enriched at GBP-targeted vacuoles", which implies that iNOS is enriched on GBP-targeted vacuoles compared to those not targeted by GBPs. The readers can be easily misled, especially because Figure 1 already alludes to the heterogeneity in the parasite vacuoles formed within host cells previously exposed to the cytokine. However, the text and the data presented indicate that the autoSTOMP data did not show differential associated of iNOS with GBP+ vs. GBP- vacuoles. Instead of showing comparisons only with untreated samples, Figure 2 should include this important negative result, to avert any misunderstanding. Most likely, the positive identification of iNOS reflects the escalating levels of cell expression in the IFN-gamma and combination treatment conditions.
3. As evident in Figure 1 and in prior publications, the toxoplasma-cidal effects of GBPs expressed prior to parasite invasion are rapid with half-maximal killing observed 2 hrs post infection and essentially completed by 4hrs. Thus, it was surprising that the nitration assays were performed after 6hrs, when most of the killing events have already been completed.
4. The notion that macrophage activation induces "collapse" of the parasite elaborated intravacuolar network (IVN) is highly intriguing. However, the term implies that the IVN network was first formed and then later "collapsed" by effectors. Rather than inducing a collapse of the IVN, it is more likely that the IVN fails to be elaborated starting at earlier time points and throughout the invasion and infection processes. Rather than measuring cell division (which reflects static rather than cidal activities), direct measurement of vacuole disruption and plasma membrane disruption at early time points should be made to demonstrate the proposed synergy between iNOS and GBPs in clearance. Using an auxotrophic strain and using conditions that prevent cell division would also help focus the endpoints on parasite destruction events, rather than on mechanisms that prevent replication.

RESPONSE TO REVIEWERS' COMMENTS

Reviewer #1 (Remarks to the Author):

I am a little concerned that the PV might be too small for the STOMP technique. The literature values for PV appear to be 1-3 microns, that is at the functional cut-off of the STOMP technique. The authors can conduct control experiments to show that they have resolution in the single micron range. The original STOMP technique utilized a dark control, which was treated the same as the sample except the photo-irradiation was omitted. Such control experiments are necessary to convince the reader that specific proteins from the desired subcellular structure is being selected.

We share the reviewer's interest in rigorously assessing the resolution and specificity of STOMP biotin-targeting. The reviewer makes great points that caused us to realize we over-relied on citing previous methodologies and controls in the presentation of the data.

A dark control was included in this experiment, and we have modified table 2 and the materials and methods to clarify this. 256 proteins were identified in at least 1 of 3 dark control replicates. Most of these proteins had extremely low LFQ values and were not detected in all dark controls. To use them as exclusion criteria, we imputed the missing values for these 256 proteins. Then calculated enrichment p-value for each STOMP sample relative to the dark controls (Table 1 column X-AC). Of the 256 proteins identified in dark controls only 52 proteins were not significantly enriched (similarly expressed) in one or more STOMP samples relative to the dark controls. Most were highly abundant ribosome, histone, actin, or keratin proteins. GBP2 was detected in the dark control, however, it was similar in LFQ value to unstimulated and PAM stimulated conditions, indicating that this is background level. iNOS and other GBPs were not detected in the dark controls.

The authors can also show that their targeting is accurate by providing images before and after photo-irradiation showing that only the PV structure was targeted.

We take great care to validate STOMP targeting by 1) evaluating photobleaching of the target region, 2) re-staining coverslips with streptavidin-594 to validate the specificity of targeting, and 3) the absence of background signal biotinylation in the fields of view where STOMP targeting occurred relative to the non-targeted fields of view (due to light scatter). We have published these controls in association with the development of each new macro (Yin AC 2019), (Yin JPR 2020). The data in this manuscript was collected in direct succession to the proof of principle study Yin 2019. The inclusion of the photolabeling controls in this paper would be redundant in this manuscript, however, we have included them here for the referees. We have also described and cited this control in the materials and methods of this report.

Reviewer Fig 1. (from Yin et al. 2019)

Finally, a control experiment showing that STOMP analysis of a different part of the cell (e.g. nucleus or cytoplasm) produces a different set of proteins would also provide necessary reassurance to the reader.

This is a great point that we have been working on for a while. Frustratingly, this has been more complicated than anticipated. First, we've found that optimal peptide recovery from nucleic acid-rich areas of the cell (including nuclei and stress granules) requires a different biochemical purification procedure, which precludes direct comparison to the data set shown here. Similar to the proposed 'cytoplasm' control, we settled on a 'dilate' control where the vacuole target region is dilated by one pixel and the area adjacent to the vacuole target region is photo-biotinylated to acquire a 'near neighbor' region to the vacuole (Supp. Table 1, Extended data Fig 2). We reasoned that targeting the control in this way had the potential to provide additional information about the resolution of the technique. In hindsight, this may have been overly ambitious as a control for the specificity of targeting, but it has proven useful to directly address reviewer 1 and reviewer 3's questions about resolution. Please note that it can take up to 3 days to photo-biotinylate sufficient samples for a single replicate, so performing dilate controls for all 6 sample conditions in triplicate was beyond the turnaround time for this revision. Moreover, we have been advised by our core facility director that there are likely technical error associated with Velos Orbitrap maintenance that will increase noise when comparing samples collected after so much time has elapsed. Despite these caveats, we have tried our best to address the reviewer's concerns using MAXquant LFQ to normalize this source of variability between the IFN γ samples previously acquired and this new control.

As expected, iNOS (abundant in the cytosol) and the IRGs/GBPs (stored on vesicles, organelles including Golgi and ER and regulate mitochondrial turnover) were abundant in the dilate control. These proteins were not significantly differentially enriched between dilate and all PVM samples or the GBP2+ samples, although there was a shift towards relative enrichment in the GBP2+ regions, most notably for iNOS (Extended Data Fig 2b vs. 2a relative enrichment). Another useful piece of information from this control was that parasite proteins were detected in both samples; while other landmarks, including sorting nexins (snx1, -4 and -8) which interact with PtdIns(3) and PtdIns(3,5)P2 and regulate SNARE complexes previously shown to be enriched at the vacuole (Cygan et al 2020) as well as PyCARD (ASC) the inflammasome adaptor

(Gorfu 2014, Ewald 2014, Fisch 2019) were enriched on the parasite vacuole or GBP2+ samples. Cumulatively, these data indicate that the resolution of autoSTOMP is closer to 2 μ m (the PVM and dilate control regions combined) and that we are likely detecting some PV proteins as well as highly upregulated proteins present in the near neighbor region. We have gone through the text to make sure that we are not over-interpreting the data. This is detailed in the results section at lines 136-147:

"Few proteins were significantly differentially enriched when PVM and GBP2+ samples were compared within the 'IFN γ ' or the 'IFN γ and Pam3CSK4' stimulated conditions (Fig. 2e). Since the resolution of autoSTOMP (~1 μ m) wider than the parasite vacuole membrane structure, we sought to determine if sampling of near neighbor regions of the cell could be contributing to this signature by dilating the PVM Map by an additional pixel (~1 μ m) and identifying proteins in this region (Supplementary Table 1, Extended Data Fig. 2). Although there was a shift towards greater enrichment of GBPs and IRGs in the IFN γ -GBP2+ PVM vs. dilate sample compared to IFN γ -PVM, they were not significantly differentially enriched. Notably, both samples contained some significantly differentially enriched proteins. Thus we interpret these data to mean that the resolution of autoSTOMP in these assays is ~2 μ m and that the technique is identifying a mixture of vacuole-associated proteins and high abundance near neighbor proteins in the infected cell."

This is also noted in reference to iNOS at line 152:

"iNOS was also enriched in the IFN γ -stimulated GBP2+ region relative to the unprimed condition and the IFN γ PVM condition, although this was not statistically significant due to variation in the replicates, and potentially, abundant expression in the cytosol (Fig. 2d, 2e)."

While we agree with the author that the target region is almost certainly picking up proteins that do not directly interact with the PV/IVN, it is important to note that we are *not* claiming to publish comprehensive vacuole proteomes. Rather, we are using AutoSTOMP as a discovery tool to identify candidates for validation in the process of parasite recognition and clearance from the cell. While we look forward to increasing the sensitivity and selectivity of this approach, we think this study shows that the current tool can be used to advance our understanding of the mechanism of IIG-mediated parasite clearance.

The link to GitHub in their methods section was not working.

We apologize for the difficulty linking to the GitHub site. The manuscript .doc file linked correctly but a line number was added to the .pdf conversion which broke the link. We have included a new link specific to the protocols used in this paper.

They also need to reference the original STOMP technique: Hadley et al., Determining composition of micron-scale protein deposits in neurodegenerative disease by spatially targeted optical microproteomics. *Elife*. 2015 Sep 29;4:e09579. doi: 10.7554/eLife.09579. PMID: 26418743; PMCID: PMC4630677.

Agreed! this is now included.

Reviewer #2 (Remarks to the Author):

Summary: In the manuscript titled “iNOS is necessary for GBP-mediated T. gondii restriction in murine macrophages via vacuole nitration and intravacuolar network collapse,” Zhao et al. use mice, murine-derived immune cells, and Type II T. gondii parasites to ask how GBPs and iNOS interact to restrict parasite growth. The manuscript includes strong evidence for a collaborative role of GBPs and iNOS to restrict T. gondii in vitro. Authors also provide new evidence that iNOS in LysM-positive myeloid cells influences T. gondii infection in vivo. Nitration as a mechanism for iNOS/NO-mediated parasite restriction is interesting. Collectively, iNOS-related experiments add depth to our knowledge of iNOS in T. gondii infection. Enriching for GBP-positive vacuoles with autoSTOMP was a clever and novel way to leverage heterogeneity in the host response to T. gondii. It revealed new relationships in a complex immune signaling pathway. Comments are largely to expand on logic and interpretation, no additional experiments required.

Thank you for this constructive feedback and support of the work. We have made several modifications to the text in line with this feedback.

Comments:

- for non-immunologists, a schematic in Figure 1 outlining the signaling pathways discussed (GBPs, IFN γ , TLRs, iNOS, STATs) will get readers up to speed more quickly

Agreed! This has now been included as Figure 1A.

- authors should state why they select GBP2 for autoSTOMP but chromosome 3 GBPs for knockout experiments. is the choice for technical or scientific reasons or both?

We have now clarified this rationale Line 97-100:

“GBP2 was selected because there is evidence that GBP targeting is downstream of p47 IRG recruitment to the vacuole, GBP2 is the homolog of human GBP1, and GBP2 has been shown to interact with both the parasite vacuole membrane and the parasites following vacuole permeabilization.”

- line 251: there is no figure 5i

We had attempted to differentiate between the subpanels of Fig. 5F as ‘5F-i’ and ‘5F-ii.’ This was confusing when we reformatted the figure callouts to the lowercase alphabet. We now refer to these as ‘Fig. 5f top panel’ and Fig 5f ‘bottom panel.’

- line 261-263: because chromosome 3 GBPs are absent in Δ Gbp-chr3 cells (not just absent from PVM), it’s possible they play additional roles: signaling, indirect effects on other GBPs, etc.

Correct, we have modified Line 269-270:

“Although it is formally possible that chromosome 3 GBPs play a signaling role distal to the vacuole, these data are consistent with the model that chromosome 3 GBP recruitment to the vacuole is necessary for optimal RNS-mediated clearance of parasites downstream of iNOS activation and synthesis of *NO.”

- line 282: should be ‘transmission’ not ‘transmitted’
This has been amended.

- line 311, arginine discussion: evidence for nitration is good. Evidence against arginine depletion is ok (2mM arginine should guarantee parasites are replete with arginine), but not perfect. Arginine dynamics are controlled by multiple host and parasite transporters, enzymes like arginase that can consume huge quantities of arginine, and restricted exchange of subcellular arginine pools. The claim would be strengthened by definitively measuring intracellular arginine or even intraparasitic arginine, but for this paper that is probably excessive. Rather, recommend noting that arginine supplementation experiments do not definitively exclude arginine depletion.

As suggested, we have altered the language describing this result at line 229-231:

“While this experiment does not definitively exclude arginine depletion as a means of limiting T. gondii viability, it prompted us to explore alternative mechanisms of iNOS function.”

- if authors will pursue myeloid NOS2 further, comparison of LysM-driven Cre with Tie2-driven Cre may be warranted. Tie2 drives deletion in a more complete subset of myeloid cells (with the caveat that Tie2 also drives deletion in the endothelium)

We agree that we have only scratched the surface here. We added the following statement to the discussion at line 413-418:

“The lysM promoter drives expression on a range of bone marrow-derived myeloid cell lineages⁴⁹. While iNOS was not robustly detected in peritoneal neutrophils, we cannot exclude the contribution of neutrophils at earlier time points or other tissues. Future studies using more selective cre-drivers (eg. Tie2-cre⁷⁴, CX3CR1-cre⁷⁵, CD11C-cre⁷⁶) may inform the role of iNOS in specific macrophage, dendritic cell, or neutrophil populations in host resistance to T. gondii.”

Reviewer #3 (Remarks to the Author):

The intracellular mechanisms activated downstream of IFN-gamma signaling that mediate the destructive (toxoplasma-cidal) and growth restrictive (toxoplasma-static) activities of macrophages and non-phagocytic cells have been the subject of several decades of research. A consensus view has been that certain effectors of cell autonomous immunity are growth restrictive (iNOS and IDO for example) and others are direct cidal (GBPs and p47 GTPases) and that each effector arm can operate independently (for example PMID 19265156, not cited in References). In this manuscript, the authors propose that iNOS is essential for GBP’s anti-toxoplasma effects through nitration of the parasite vacuole and by inducing “collapse” of the intravacuolar network formed normally by invaded parasites. If fully substantiated, the work would represent a significant advance in the field. However, multiple issues in the experimental

design and data presentation need to be addressed further.

This critique has provided a valuable lens through which we have re-evaluated our data and the rigor of the literature. In addition to modifying some of our language, they caused us to design several new experiments that have greatly enhanced the quality of the study.

Before addressing the detailed critiques, it seems important to respond to the reviewer's statement that 'certain effectors of cell-autonomous immunity are growth restrictive (iNOS and IDO for example) and others are direct cidal (GBPs and p47 GTPases) and that each effector arm can operate independently.'

We agree that there is strong evidence that GBP and IRG recruitment to the vacuole is necessary to engage a parasite-cidal process. However, we do need to be careful with the assumption that the IIGs are directly *Toxoplasma* cidal. In bacterial systems, specific GBPs have been mixed with the microbe in the presence of nucleoside and shown to be *sufficient* to form bacterial out membrane vesicles and kill bacteria (PMID: 35906252). Analogous experiments showing that any of the IRGs or GBPs are sufficient to directly kill *Toxoplasma* have not been performed (We are not even sure they are possible, at least in the same way!). Interestingly, 3 independent groups have recently shown that GBPs take on a novel, elongated formation upon interaction with LPS on bacteria (bioRxiv 2023.08.01.551421, bioRxiv 2023.03.28.534355; PMID: 32510692). While it is possible that GBPs/IRGs are sufficient for parasite killing, *Toxoplasma* lacks LPS orthologs, so the GBPs and IRGs may be functioning differently on the parasite vacuole/parasite membrane or require some other co-factor for 'cidal' activity. This rationale opens the door for the alternate hypothesis explored in our study.

This comment also caused us to think about what the true measure of a host regulator of parasite 'stasis' would be. We would define a 'growth restrictive' host effector mechanism as one that causes a parasite that has infected a host cell to persist, without dividing in the vacuole compared to a host cell where that host effector mechanism is inactive. In macrophages, the signals that are required for iNOS activation (STAT and NFkB) also upregulate IRGs and GBPs. From this perspective, clearance of a parasite by the IIGs would obscure attempts to measure the growth static function of iNOS. So, it is reasonable to conclude that endpoint assays in mouse macrophages that attempted to measure iNOS function must have been confounded by the cidal contribution of IRGs and GBPs. The paper cited has some nice information about IRG targeting (we have now cited it). But it's a perfect example of how these issues weaken the data supporting the conclusion that iNOS causes parasite stasis: We now know that IRGM3KO doesn't block IRG and GBP attack efficiently, the endpoints assays used do not directly measure parasite growth, plus it not clear that PECs elicited over 7 days in iNOSKO are the same population as PECs from WT/IRGM3KO (our Fig. 2C indicates they are likely different cell types).

To hold iNOS in isolation and test the hypothesis that iNOS has a toxoplasma-static function, we would need to completely block the IRG and GBP system downstream of STAT1/NFkB. Blocking regulatory Irgms is another potential solution, though to our knowledge global blockade of the

IRG and GBP system in macrophages has not been demonstrated. Moreover, a recent study implicates Irgm1 in the broader regulation of type I IFN signaling (PMID: 32715615) which would confound conclusions. In lieu of the perfect tool, we do provide two lines of evidence that activating the iNOS/RNS arm in isolation is not toxoplasma static (or cidal!) by itself.

- In Fig. 5d we show that treating unstimulated macrophage with NO-donor NONOATE does not significantly impair parasite growth over 14 hours compared to vehicle treatment.
- In Extended Data Fig 5 we also show that tet-inducible iNOS, which produces nitrite at physiological levels, also does not result in a reduction of parasite growth.

We also provide solid evidence for cooperation:

- Without iNOS there is a significant increase in GBP2+ vacuoles containing divided parasites.
- Treating chromosome 3 GBP KO RAWs with iNOS inhibitor rescues parasite load to the same level as iNOS inhibition or alone, indicating that these pathways function together rather than additively.
- Finally, we added live imaging experiments to show that parasites can grow, egress, and shed GBP2 in the absence of iNOS activity, further supporting the significance of cooperativity between these pathways (Fig 6c and Extended Data Fig. 8).

This is a long response, but these comments have caused us to think about our work from a new perspective, for which we are grateful.

1. In this paper, a clear definition and distinction should be made between cidal and static effects/endpoints of the activated host cells. Often in the manuscript, the terms “restriction” and “clearance” are used interchangeably. This is evident in the abstract and elsewhere in the manuscript.

The reviewer is right, we had binned the processes of growth restriction (stasis) and parasite survival under the same term ‘restriction.’ As discussed above, the only assay that measures growth directly is our new live imaging data, which is supported by data measuring parasites per vacuole. We have changed the language so that we are referring to parasite ‘load,’ ‘killing’ or ‘clearance’ to make it clear that our readouts are loss of parasites rather than impaired growth.

In Figure 1 the time course is spread over a long duration of time, but the cidal activities are mostly completed within the first 4 hours of the time course as reflected by the drastic attrition of the number of vacuoles present at 14 hpi.

We wanted to dig into this more deeply because we don’t think we can attribute the early loss of luciferase signal in this bulk assay to IIG-cidal activity. For example, phagocytosis could contribute to this early loss of luciferase signal. We also see a ~25% loss of luciferase signal at 4 hours in the unstimulated condition (no iNOS or IIG expression), which reflects our typical

plaque efficiency after syringe lysis-this almost certainly accounts for a similar proportion of loss of signal observed in IFN γ primed conditions.

To directly test the rate of GBP targeting and measure parasite viability, we developed a system for live cell imaging. Using this assay, we show that BFP-tagged GBP2 can target vacuoles as early as the first imaging frame, 1-hour post-infection, and as late as 17 hours. Parasite loss of fluorescence and morphological changes consistent with loss of viability also through our 19 hours of imaging. Together these data indicate that in our system, there is a long window of GBP-dependent parasite cidal activity. The median time to parasite clearance (loss of GFP signal) is significantly lengthened and the majority of parasites remain viable when iNOS is inhibited. Interestingly, some parasites are still killed in the iNOS-inhibited condition. Assuming that these are not simply host cells with extremely high levels of iNOS that escape inhibitor treatment, we can use this experimental system to independently evaluate these two modes of clearance in future studies. For example, perhaps the GBP/IRG system is most efficient at clearing compromised parasites- whether this is related to damage to the parasite during syringe lysis (potentially explaining the early wave of targeting), parasites that express sub-optimal levels of effectors (eg Rop17/18), or that have been damaged by redox stress. It is worth noting that the live imaging experiment underreports the total number of 'viable' parasites in the culture because some of the surviving parasites have divided. This may also explain discrepancies between the luciferase assay and the live imaging assay readouts.

In anticipation of the focus on GBPs and iNOS, the expression levels of these IFN- γ induced effectors should be shown under all four conditions studies: unprimed, Pam3CSK4, IFN- γ and combination treatments.

In Figure 1 we show GBP2 recruitment in all 4 stimulation conditions and quantify parasites per vacuole. We have included the MaxQuantLFQ protein values for all IIG family members identified in all conditions of the STOMP data sets (Fig 2D, Table 1). In Figure 4 iNOS transcript and protein levels are shown in all four conditions. Although we do not have selective antibodies for all of the GBPs or IRGs, we have shown transcript expression levels for the chromosome 3 GBPs, Irgb6, and Irgb10 in Extended Data Fig. 3.

2. The statement (heading) used in the results section to describe the autoSTOMP data (Figure 2 and extended data Fig. 2) can be readily misinterpreted and thus can be misleading. It states that "iNOS is enriched at GBP-targeted vacuoles", which implies that iNOS is enriched on GBP-targeted vacuoles compared to those not targeted by GBPs. The readers can be easily misled, especially because Figure 1 already alludes to the heterogeneity in the parasite vacuoles formed within host cells previously exposed to the cytokine. However, the text and the data presented indicate that the autoSTOMP data did not show differential associated of iNOS with GBP+ vs. GBP- vacuoles. Instead of showing comparisons only with untreated samples, Figure 2 should include this important negative result, to avert any misunderstanding. Most likely, the positive identification of iNOS reflects the escalating levels of cell expression in the IFN- γ and combination treatment conditions.

We certainly did not intend to be misleading, as described in response to Reviewer 1 we have modified the language in this section based on new dilate controls. It is also worth clarifying that we are not comparing GBP+ and GBP- vacuoles. We are looking at proteins enriched around all parasites ('PVM' samples) and comparing them to proteins enriched at the 'GBP2+' regions of the targeted vacuoles. This is why we suggest that the signal on the 'PVM' samples could be coming from the nested subpopulation that are positive for GBPs.

We have changed the title of the section to: '**iNOS is enriched near parasite vacuoles in IFN γ -treated BM-MoDCs by autoSTOMP**' and moved Supp. Fig 2 to the main body of Figure 2e. Please note that this title was shortened in keeping with the Nature Communication guidelines for section subtitle length.

3. As evident in Figure 1 and in prior publications, the toxoplasma-cidal effects of GBPs expressed prior to parasite invasion are rapid with half-maximal killing observed 2 hrs post infection and essentially completed by 4hrs. Thus, it was surprising that the nitration assays were performed after 6hrs, when most of the killing events have already been completed.

We would love to know which papers the reviewer is referencing! There is considerable variability in the kinetics and efficiency of IRG/GBP localization and parasite clearance depending on cell type and stimulation condition. As stated above, we have addressed this kinetics concern with new live imaging experiments. In addition, we have expanded Fig. 5G to show similar nitration is also observed at 2 hours post-infection as requested by the reviewer (Fig. 6c).

4. The notion that macrophage activation induces "collapse" of the parasite elaborated intravacuolar network (IVN) is highly intriguing. However, the term implies that the IVN network was first formed and then later "collapsed" by effectors. Rather than inducing a collapse of the IVN, it is more likely that the IVN fails to be elaborated starting at earlier time points and throughout the invasion and infection processes. Rather than measuring cell division (which reflects static rather than cidal activities), direct measurement of vacuole disruption and plasma membrane disruption at early time points should be made to demonstrate the proposed synergy between iNOS and GBPs in clearance. Using an auxotrophic strain and using conditions that prevent cell division would also help focus the endpoints on parasite destruction events, rather than on mechanisms that prevent replication.

This is a terrific point. We designed an experiment to distinguish between these possibilities that iNOS could *only* prevent IFN development vs. induce collapse of a formed IVN. We infected IFN γ -primed iNOS KO macrophages, allowed the infection to proceed for 6 hours, and then spiked in a nitric oxide donor for 4 hours before collecting samples for TEM (Extended Data Fig. 9b). Spiking in NONOate was sufficient to reduce the IVN volume, consistent with bonified collapse. We also paired this experiment with a luciferase assay to confirm that spiking in NONOate was sufficient to rescue parasite killing in the iNOS KO cells at later endpoints (Extended Data Fig. 9c). The TEM data paired with our live cell imaging data using iNOS inhibitors suggested that the function of iNOS and IVN collapse does not only affect parasite

replication. Inhibiting iNOS also increased the incidence of mTagBFP2-GBP2+ parasite egress, shedding of GBP2, and replication, consistent with our model that iNOS cooperates with GBPs for optimal parasite killing.

REVIEWERS' COMMENTS

Reviewer #1 (Remarks to the Author):

In their re-submission, the authors clarification that they were using the STOMP technique as a targeted method rather than a proteomic study of parasitic vesicles, made me look at the manuscript differently. I accept the authors discussion of the other controls I suggested. Their estimation of the resolution of their setup being 2 microns was appreciated, which provided further reassurance to the reader that at this resolution level their system is capable of identifying targets for validation.

Reviewer #2 (Remarks to the Author):

The authors have done an excellent job addressing all of the concerns from the previous reviews. This is a highly significant manuscript for the parasitology field.

Reviewer #3 (Remarks to the Author):

The authors have thoroughly and thoughtfully, in the revised submission, addressed the key questions and concerns raised in the prior review of this manuscript.